# ShotBench: Expert-Level Cinematic Understanding in Vision-Language Models

**Hongbo Liu**[1,2*]  **Jingwen He**[3,2*]  **Yi Jin**[1]  **Dian Zheng**[2]
**Yuhao Dong**[4]  **Fan Zhang**[2]  **Ziqi Huang**[4]  **Yinan He**[2]
**Weichao Chen**[1]  **Yu Qiao**[2]  **Wanli Ouyang**[3,2]  **Shengjie Zhao**[1†]  **Ziwei Liu**[4†]
[1]Tongji University, [2]Shanghai Artificial Intelligence Laboratory,
[3]The Chinese University of Hong Kong, [4]S-Lab, Nanyang Technological University
[*]Equal contribution.    [†]Corresponding authors.

**Project Page:** `https://vchitect.github.io/ShotBench-project/`

## Abstract

Cinematography, the fundamental visual language of film, is essential for conveying narrative, emotion, and aesthetic quality. While recent Vision-Language Models (VLMs) demonstrate strong general visual understanding, their proficiency in comprehending the nuanced cinematic grammar embedded within individual shots remains largely unexplored and lacks robust evaluation. This critical gap limits both fine-grained visual comprehension and the precision of AI-assisted video generation. To address this, we introduce **ShotBench**, a comprehensive benchmark specifically designed for cinematic language understanding. It features over 3.5k expert-annotated QA pairs from images and video clips, meticulously curated from over 200 acclaimed (predominantly Oscar-nominated) films and spanning eight key cinematography dimensions. Our evaluation of 24 leading VLMs on ShotBench reveals their substantial limitations: even the top-performing model achieves less than 60% average accuracy, particularly struggling with fine-grained visual cues and complex spatial reasoning. To catalyze advancement in this domain, we construct **ShotQA**, a large-scale multimodal dataset comprising approximately 70k cinematic QA pairs. Leveraging ShotQA, we develop **ShotVL** through supervised fine-tuning and Group Relative Policy Optimization. ShotVL significantly outperforms all existing open-source and proprietary models on ShotBench, establishing new **state-of-the-art** performance. We open-source our models, data, and code to foster rapid progress in this crucial area of AI-driven cinematic understanding and generation.

## 1  Introduction

Cinematography, the art of crafting visual narratives through meticulously designed shots [4, 17], forms the bedrock of high-quality filmmaking. Each shot, from framing and lens choice to lighting and camera movement, is deliberately composed to convey narrative meaning, emotional tone, and aesthetic impact. For text-to-image/video generation [2, 11, 23, 24, 40, 59] to achieve similar cinematic quality, it requires a mechanism capable of understanding these cinematographic principles. Vision-Language Models (VLMs) [3, 26, 33, 35, 48, 61, 65] are the primary candidates for developing such understanding. Thus, the core challenge is whether current VLMs can genuinely grasp the nuanced language of cinematography and its artistic intent, moving beyond literal scene interpretation. This deep cinematographic comprehension remains significantly underexplored. Existing VLM benchmarks, while diverse [8, 21, 31, 63], typically lack the necessary focus for robust cinematographic evaluation, a gap exacerbated by a scarcity of specialized models, datasets with rich cinematic annotations, and consequently, rigorous benchmarks for this specific type of understanding.

To bridge this critical gap, we introduce **ShotBench**, a comprehensive benchmark specifically designed to assess VLMs' understanding of cinematic language. ShotBench comprises over 3.5k

39th Conference on Neural Information Processing Systems (NeurIPS 2025).

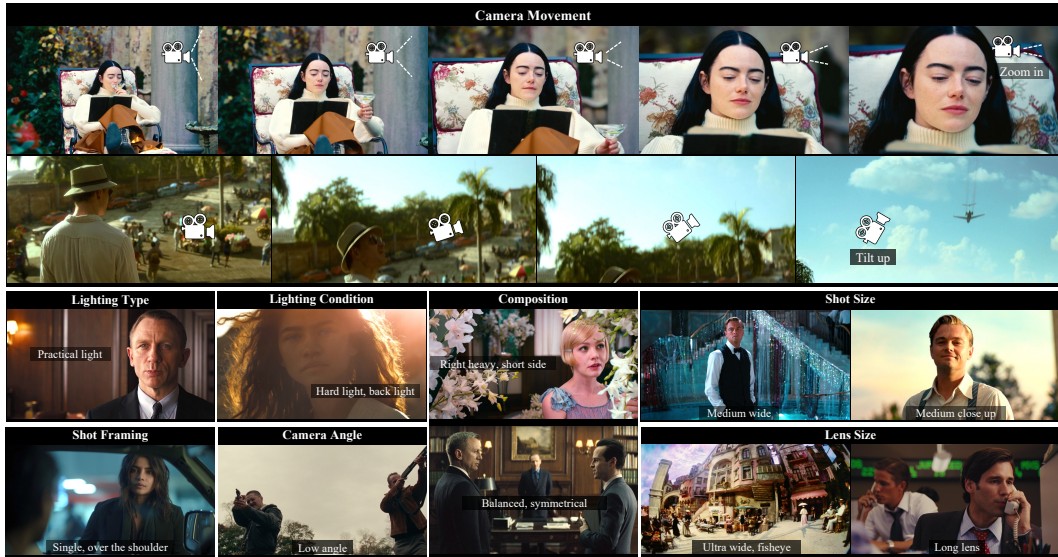

Figure 1: **Overview of ShotBench**. The benchmark covers eight core dimensions of cinematography: *shot size*, *framing*, *camera angle*, *lens size*, *lighting type*, *lighting condition*, *composition*, and *camera movement*.

expert-annotated multiple-choice QA examples, meticulously curated from both images and video clips across over 200 films, predominantly those that have received Oscar nominations for Best Cinematography[1]. It rigorously spans eight fundamental cinematography dimensions: *shot size*, *shot framing*, *camera angle*, *lens size*, *lighting type*, *lighting condition*, *composition*, and *camera movement*. Our rigorous annotation pipeline, combining trained annotators with expert oversight, ensures a high-quality evaluation set grounded in professional cinematic knowledge.

We conduct an extensive evaluation of 24 leading open-source and proprietary VLMs on ShotBench. Our results reveal that even the strongest VLM (GPT-4o [38]) in our evaluation averages below 60% accuracy, clearly indicating a considerable gap between current VLM capabilities and genuine cinematographic comprehension. In-depth analysis further highlights specific weaknesses: advanced models such as GPT-4o [38] and Qwen2.5-VL [3], despite grasping core cinematic concepts, often struggle to map subtle visual details to precise professional terminology (e.g., distinguishing a `medium shot` from a `medium close-up`). They also demonstrate constrained spatial reasoning, especially regarding camera position and angle. Strikingly, the camera movement dimension proved exceptionally challenging, with over half of the models failing to surpass 40% accuracy.

To further advance cinematography understanding in VLMs, we construct **ShotQA**, the first large-scale multimodal dataset for cinematic language understanding, consisting of approximately 70k high-quality QA pairs derived from movie images and video clips. Leveraging ShotQA, we develop **ShotVL**, an optimized VLM series based on Qwen2.5-VL-3B and Qwen2.5-VL-7B [3], trained with supervised fine-tuning and Group Relative Policy Optimization (GRPO) [46] to enhance its alignment of visual features with cinematography knowledge and strengthen its reasoning capabilities. Experimental results demonstrate that ShotVL achieves consistent and substantial improvements across all ShotBench dimensions, establishing new **state-of-the-art** performance and decisively surpassing both the best-performing open-source (Qwen2.5-VL-72B-Instruct [3]) and proprietary (GPT-4o [38]) models.

Our contributions are summarized as follows:

- We introduce **ShotBench**, a comprehensive benchmark for evaluating VLMs' understanding of cinematic language. It comprises over 3.5k expert-annotated QA pairs derived from images and video clips of over 200 critically acclaimed films (predominantly Oscar-nominated), covering eight distinct cinematography dimensions. This provides a rigorous new standard for assessing fine-grained visual comprehension in film.

---

[1]https://en.wikipedia.org/wiki/Academy_Award_for_Best_Cinematography

- We conducted an extensive evaluation of 24 leading VLMs, including prominent open-source and proprietary models, on ShotBench. Our results reveal a critical performance gap: even the most capable model, GPT-4o, achieves less than 60% average accuracy. This systematically quantifies the current limitations of VLMs in genuine cinematographic comprehension.
- To address the identified limitations and facilitate future research, we constructed **ShotQA**, the first large-scale multimodal dataset for cinematography understanding, containing approximately 70k high-quality QA pairs. Leveraging ShotQA, we developed **ShotVL**, a novel VLM series trained using Supervised Fine-Tuning (SFT) and Group Relative Policy Optimization (GRPO). ShotVL significantly surpasses all tested open-source and proprietary models, establishing a new state-of-the-art on ShotBench.

## 2 Related Work

### 2.1 Benchmarking Vision-Language Models

Vision-Language Models (VLMs) [3, 26, 33, 35, 48, 61, 65] are large-scale models designed to integrate visual perception with natural language understanding. In recent years, VLMs have demonstrated strong capabilities across perception, reasoning, and a wide range of multi-disciplinary applications [1, 15, 27, 30, 31, 34, 49, 56, 57]. Recently, researchers have proposed a variety of benchmarks to assess VLMs' capability. For example, MMBench [31] evaluates VLMs across 20 distinct ability dimensions, and MMVU [63] focuses on video understanding across four core academic disciplines. Other benchmarks target specific cognitive or reasoning capacities: LogicVista [54] assesses visual logical reasoning in a multi-choice format, and SPACE [41] systematically compare spatial reasoning abilities between VLMs and animals. Additional efforts, EgoSchema [37], and VSI-Bench [55], evaluate egocentric video understanding. Moreover, some works introduce tasks with specific domains, such as scientific and mathematical figure interpretation [44, 53], knowledge acquisition [21], and visual coding [60].

### 2.2 Cinematography Understanding

Early work on automatic film analysis includes many sub-tasks such as shot type classification [42], scene segmentation [43, 47, 58], and cut type recognition [39]. For example, MovieShots [42] categories shots into five scale types and four camera movements types, providing an early taxonomy for cinematography understanding. With the rapid progress in image and video generation, film-level generation has begun to attract increasing attention [11, 23, 24, 40, 59]. Many recent works rely on VLMs to synthesise large training corpora [23, 59]. However, they often introduce additional classifiers to identify camera movements or shot sizes. For example, HunyuanVideo [23] trains a camera movement classifier capable of predicting 14 distinct camera movement types, introducing additional training and data annotation overhead.

Table 1: Cinematography Understanding Benchmark Comparison

| Dimensions | MovieShots [42] | MovieNet [22] | CineScale2 [45] | CameraBench [29] | CineTechBench [52] | ShotBench |
|---|---|---|---|---|---|---|
| *Shot size* | ✔ | ✔ | ✗ | ✗ | ✔ | ✔ |
| *Shot framing* | ✗ | ✗ | ✗ | ✗ | ✗ | ✔ |
| *Camera angle* | ✗ | ✗ | ✔ | ✗ | ✔ | ✔ |
| *Lens size* | ✗ | ✗ | ✗ | ✗ | ✔ | ✔ |
| *Lighting type* | ✗ | ✗ | ✗ | ✗ | ✗ | ✔ |
| *Lighting condition* | ✗ | ✗ | ✗ | ✗ | ✔ | ✔ |
| *Composition* | ✗ | ✗ | ✗ | ✗ | ✔ | ✔ |
| *Camera movement* | ✔ | ✔ | ✗ | ✔ | ✔ | ✔ |

## 3 ShotBench

To evaluate the capabilities of VLMs on cinematography understanding, we first define the concept of cinematography understanding and introduce ShotBench in 3.1. Next, we provide a detailed description of the data collection process in 3.2. Using ShotBench, we then perform evaluations to assess whether VLMs can effectively comprehend cinematic conventions and analyze potential causes of their performance limitations in 3.3.

## 3.1 Overview

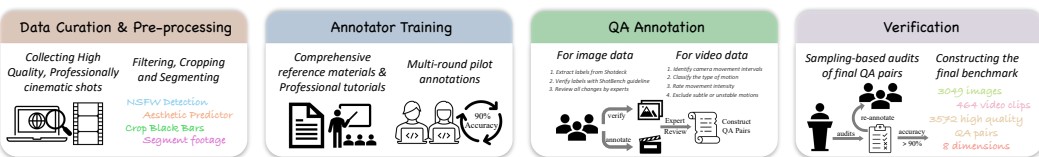

Figure 2: An overview of the ShotBench construction pipeline.

Understanding cinematography involves not only identifying visual elements like framing, lighting, and camera movement, but also interpreting how they work together to convey narrative and mood. While recent VLMs show some ability to recognize cinematic language, their deeper understanding of cinematic conventions remains underexplored. Here, we introduce ShotBench, a dedicated benchmark designed to evaluate VLMs' understanding of cinematography language in a comprehensive and structured manner. ShotBench covers eight core dimensions [2] commonly used in cinematic analysis: *shot size*, *shot framing*, *camera angle*, *lens size*, *lighting type*, *lighting conditions*, *composition*, and *camera movement*. These dimensions reflect key principles of visual storytelling in film production and serve as the foundation for evaluating model comprehension.

Each sample in ShotBench is paired with a multiple-choice question targeting a specific cinematography aspect, requiring the model to not only perceive the scene holistically, but also extract fine-grained visual cues to reason about the underlying cinematic techniques. An overview of the benchmark framework is illustrated in Figure 1.

## 3.2 Data Construction Process

To construct ShotBench, we design a systematic data collection and processing pipeline, as illustrated in Figure 2. The process consists of four key stages: Data Curation & Pre-processing, Annotator Training, QA Annotation, and Verification.

**Data Curation & Pre-processing**   We collect the dataset primarily from films that won or were nominated for the Academy Award for Best Cinematography, ensuring high-quality and professionally crafted shot. Data are sourced from public websites and include high-resolution images and video clips. To ensure quality and safety, we apply the LAION aesthetic predictor [25] for filtering low-quality samples, NSFW detection [50] to remove inappropriate content, and FFmpeg [18] to crop black bars. For video processing, we use TransNetV2 [47] to segment footage into individual shots. The full list of collected movies is provided in Appendix C.

**Annotator Training**   To ensure high-quality annotations, we first curated comprehensive reference materials from publicly available cinematography tutorials covering all eight dimensions in ShotBench. Annotators were required to study these materials before labeling. We then conducted multi-round pilot annotations, supported by expert audits and daily discussions to resolve ambiguities. All issues and resolutions were documented to guide the final annotation phase.

**QA Annotation**   Based on ShotBench's predefined dimensions, we automatically generated question prompts using templated formats (e.g., *"What is the shot size of this movie shot?"*). We ensured an even distribution of questions across the eight dimensions, as illustrated in Appendix C (Figure 15b). For image data, we extracted candidate labels from Shotdeck [3], a professional cinematography reference platform, where metadata had been curated by experienced photographers. Annotators verified these labels against ShotBench guidelines and corrected any discrepancies. All label modifications were reviewed by experts. For videos, annotators identified all valid camera movement intervals by marking start and end timestamps.

**Verification**   All question–answer pairs were reviewed through multiple expert audits, with batches revised iteratively until reaching satisfactory quality. Through this rigorous pipeline, we further sampled from the validated data to construct the final benchmark, consisting of 3,049 images and

---

[2]https://en.wikipedia.org/wiki/Cinematography
[3]https://shotdeck.com

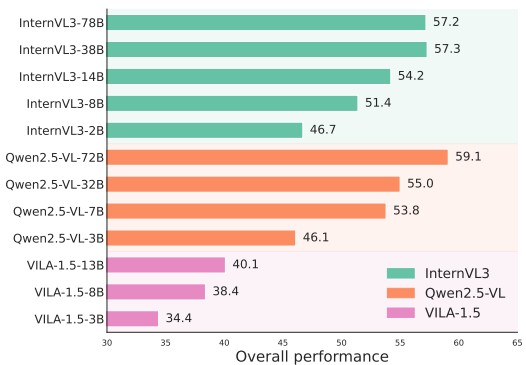
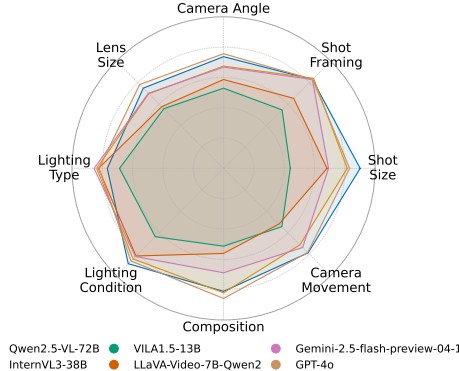

Figure 3: Overall performance comparison of InternVL3, Qwen2.5-VL, and VILA-1.5 model families, highlighting variations by model size. The results consistently show that larger models within each series generally yield superior performance outcomes.

Figure 4: Performance evaluation of six Vision Language Models (VLMs) on cinematographic understanding, visualized across several dimensions. Stronger models perform well uniformly, without specific dimensional weaknesses.

464 video clips, resulting in 3,572 high-quality question-answer pairs across all eight ShotBench dimensions.

## 3.3 Evaluation

**Setup**  To provide a comprehensive assessment of the challenges posed by ShotBench and establish reference baselines for future research, we evaluate a diverse set of state-of-the-art multimodal foundation models that support video or multi-image inputs. Specifically, we evaluate a total of 24 foundation models, including both open-source and proprietary models: Qwen2.5-VL [3], LLaVA-Video [62], LLaVA-OneVision [26], InternVL-2.5 & 3 [7, 65], InternLM-XComposer-2.0 [14], Ovis2 [36], VILA1.5 [28], InstructBLIP [9], and Gemini-2.0 & 2.5 [12, 13]. All ShotBench questions are designed as four-option single-choice questions. For questions involving multiple keywords (e.g., lighting condition), each option contains the same number of keywords to maintain balance. Only the correct option includes all the correct keywords, while the distractors may contain a mixture of correct and incorrect keywords to enhance the challenge and maintain fairness. Specifically, To ensure fairness and reproducibility, we adopt the VLMEvalKit [16] framework for standardized evaluation. We report accuracy as the primary metric to quantify model performance on ShotBench. Additional implementation details and evaluation prompts are provided in the Appendix B.

**Results and Findings**  The evaluation results are reported in Table 2, yield several key findings: (1) Approximately half of the evaluated models attain an overall accuracy below 50%. Even the leading model, GPT-4o, fails to reach 60% accuracy, underscoring the significant gap between current VLMs and a true understanding of cinematography. (2) The overall performance differences between open-source and proprietary models are marginal. Notably, Qwen2.5-VL-72B-Instruct (59.1%) achieves almost the same performance as GPT-4o (59.3%) (3) The camera movement dimension represents a particular area of weakness across current models, with achieved accuracy often approximating random selection (around 25%). (4) Within each series, larger models generally achieve higher accuracy (as shown in Figure 3), suggesting a potential scaling effect with respect to model size in cinematography language understanding.

To better understand the limitations of current VLMs in cinematic language understanding, we conduct extensive quantitative and qualitative analyses on the prediction results of representative models. Our analysis reveals significant challenges for current models across three core aspects: (1) *fine-grained visual-terminology alignment*, (2) *spatial perception of camera position and orientation*, and (3) *visual reasoning in cinematography*.

**Fine-Grained Visual–Terminology Alignment**  Through extensive case studies, we find that current VLMs frequently fail to precisely align visual cues with specific cinematic terms, particularly when the task requires expert-level distinctions. Such shortcomings are especially evident in dimensions like *shot size* and *lens size*, where categories are defined by fine-grained framing or focal length conventions. For example, a `Medium Wide Shot (MWS)` typically frames the subject from

Table 2: **Evaluation results for 24 VLMs.** Abbreviations adopted: SS for *Shot Size*; SF for *Shot Framing*; CA for *Camera Angle*; LS for *Lens Size*; LT for *Lighting Type*; LC for *Lighting Conditions*; SC for *Shot Composition*; CM for *Camera Movement*. **Bold** indicates the best result, and underline indicates the second best in each group.

| Models | SS | SF | CA | LS | LT | LC | SC | CM | Avg |
|---|---|---|---|---|---|---|---|---|---|
| *Open-Sourced VLMs* | | | | | | | | | |
| Qwen2.5-VL-3B-Instruct [3] | 54.6 | 56.6 | 43.1 | 36.6 | 59.3 | 45.1 | 41.5 | 31.9 | 46.1 |
| Qwen2.5-VL-7B-Instruct [3] | 69.1 | 73.5 | 53.2 | 47.0 | 60.5 | 47.4 | 49.9 | 30.2 | 53.8 |
| LLaVA-NeXT-Video-7B [62] | 35.9 | 37.1 | 32.5 | 27.8 | 50.9 | 31.7 | 28.0 | 34.8 | 34.4 |
| LLaVA-Video-7B-Qwen2 [62] | 56.9 | 65.4 | 45.1 | 36.0 | 63.5 | 45.4 | 37.4 | 35.3 | 48.1 |
| LLaVA-Onevision-Qwen2-7B-Ov-Chat [26] | 58.4 | 71.0 | 52.3 | 38.7 | 59.5 | 44.9 | 50.9 | 39.7 | 51.9 |
| InternVL2.5-8B [7] | 56.3 | 70.3 | 50.8 | 41.1 | 60.2 | 45.1 | 50.1 | 33.6 | 50.9 |
| InternVL3-2B [65] | 56.3 | 56.0 | 44.4 | 34.6 | 56.8 | 44.6 | 43.0 | 38.1 | 46.7 |
| InternVL3-8B [65] | 62.1 | 65.8 | 46.8 | 42.9 | 58.0 | 44.3 | 46.8 | 44.2 | 51.4 |
| InternVL3-14B [65] | 59.6 | 82.2 | 55.4 | 40.7 | 61.7 | 44.6 | 51.1 | 38.2 | 54.2 |
| Internlm-xcomposer2d5-7B [14] | 51.1 | 71.0 | 39.8 | 32.7 | 59.3 | 35.7 | 35.7 | 38.8 | 45.5 |
| Ovis2-8B [36] | 35.9 | 37.1 | 32.5 | 27.8 | 50.9 | 31.7 | 28.0 | 31.3 | 34.9 |
| VILA1.5-3B [28] | 33.4 | 44.9 | 32.1 | 28.6 | 50.6 | 35.7 | 28.4 | 21.5 | 34.4 |
| VILA1.5-8B [28] | 40.6 | 44.5 | 39.1 | 29.7 | 48.9 | 32.9 | 34.4 | 36.9 | 38.4 |
| VILA1.5-13B [28] | 36.7 | 54.6 | 40.7 | 34.8 | 52.8 | 35.4 | 34.2 | 31.3 | 40.1 |
| Instructblip-vicuna-7B [9] | 27.0 | 27.9 | 34.5 | 29.4 | 44.4 | 29.7 | 27.1 | 25.0 | 30.6 |
| Instructblip-vicuna-13B [9] | 26.8 | 29.2 | 27.9 | 28.0 | 39.0 | 24.0 | 27.1 | 22.0 | 28.0 |
| InternVL2.5-38B [7] | 67.8 | **85.4** | 55.4 | 41.7 | 61.7 | 48.9 | 52.4 | 44.0 | 57.2 |
| InternVL3-38B [65] | 68.0 | 84.0 | 51.9 | 43.6 | 64.4 | 46.9 | 54.7 | 44.6 | 57.3 |
| Qwen2.5-VL-32B-Instruct [3] | 62.3 | 76.6 | 51.0 | 48.3 | 61.7 | 44.0 | 52.2 | 43.8 | 55.0 |
| Qwen2.5-VL-72B-Instruct [3] | **75.1** | 82.9 | 56.7 | 46.8 | 59.0 | **49.4** | 54.1 | **48.9** | 59.1 |
| InternVL3-78B [65] | 69.7 | 80.0 | 54.5 | 44.0 | **65.5** | 47.4 | 51.8 | 44.4 | 57.2 |
| *Proprietary VLMs* | | | | | | | | | |
| Gemini-2.0-flash [12] | 48.9 | 75.5 | 44.6 | 31.9 | 62.2 | 48.9 | 52.4 | 47.4 | 51.5 |
| Gemini-2.5-flash-preview-04-17 [13] | 57.7 | 82.9 | 51.4 | 43.8 | 65.2 | 45.7 | 45.9 | 43.5 | 54.5 |
| GPT-4o [38] | 69.3 | 83.1 | **58.2** | **48.9** | 63.2 | 48.0 | **55.2** | 48.3 | **59.3** |

Figure 5: Confusion matrices of GPT-4o' predictions on **shot size** (left) and **lens size** (right).

the knees up, while a `Medium Shot (MS)` frames from the waist up. Regarding *lens size*, `Ultra Wide` offer a broader field of view and often introduce edge distortion, whereas `Long Lens` compress spatial depth, making the foreground and background elements appear closer. We draw the confusion matrices based on results of GPT-4o, shown in Figure 5. It reveals that most misclassifications occur between visually adjacent categories. For instance, `MS` is frequently confused with `MCU` (36.2%) or `MWS` (10.1%), and `Medium lens` is often misclassified as `Wide` or `Long lens`.

These findings suggest that current VLMs lack fine-grained alignment needed to reliably distinguish between visually similar but semantically distinct categories. A plausible explanation is that the training data used for these models may lack sufficient annotation granularity or consistency in cinematography labeling, limiting their ability to internalize professional-level distinctions.

**Spatial Perception of Camera Position and Orientation**   ShotBench systematically evaluates this ability at the level of cinematic language, covering concepts such as camera angle, position, and focal

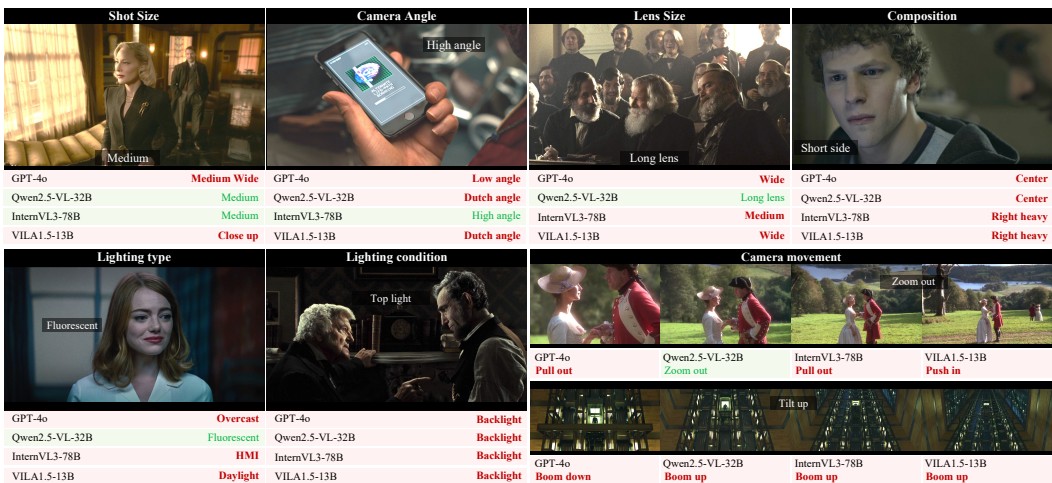

Figure 6: Examples of failure cases where VLMs struggle with fine-grained visual–terminology alignment, spatial perception, and visual reasoning.

length changes. ShotBench evaluates both static and dynamic camera attributes. For static scenarios, models are tested on fixed-angle concepts such as `low angle` and `high angle`. For dynamic cases, the camera movement dimension probes a model's ability to recognize changes in position (e.g., `pull out`), angle (e.g., `tilt up`), and focal length (e.g., `zoom in`). Results show that even the best-performing model, GPT-4o, achieves only 58.2% accuracy on static camera angle recognition, indicating its struggle in perceiving and reasoning the camera orientation in the space. The situation is worse for camera movements, where more than half of the evaluated models fall below 40% - substantially lower than their performance on other ShotBench dimensions.

Our case study analysis reveals that frontier models often show solid textual understanding of camera-related terms. However, they often fail to make correct predictions in practice. For instance, almost no existing model successfully distinguishes between position change (`push in`) and focal length change (`zoom in`), a task that requires perceiving parallax ( 6 (second row, third column). Besides, even identifying a `high angle` may result in incorrect predictions due to misperception of camera height and tilt (Figure 6, top row, second case), not to mention the change of orientation ( 6 (second row, third column).

**Visual Reasoning in Cinematography**    We observe that understanding some dimensions might need VLM to reason like a cinematography expert. For example, recognizing a short side composition (Figure 6, second row, third column) requires the model to infer the subject's gaze direction relative to their frame position—a subtle yet important cue. Similarly, identifying fluorescent lighting may involve reasoning the light source based on the subject's color tone, apparent color temperature, and the direction and softness of shadows in the scene (Figure 6, second row, first column). We hypothesize that reasoning processes can help VLMs attend to critical visual details relevant to cinematic semantics—such as spatial reasoning for determining camera angle or lens size, identifying camera movement from the motion of elements within the frame, and even discerning the director's intent in guiding the viewer's attention through compositional choices. We provide quantitative and qualitative analyses in Section 5.3 and Appendix A.2. Our findings suggest that encouraging VLMs to engage in structured reasoning provides noticeable improvements in their ability to understand cinematic language.

# 4    ShotQA & ShotVL: Advancing Cinematography Understanding via Targeted Training

To address the nuanced challenge of enabling Visual Language Models (VLMs) to perceive and reason about cinematic elements, we introduce **ShotQA**, a novel large-scale dataset, and **ShotVL**, a VLM series specifically designed for cinematography understanding. ShotVL employs a strategic two-stage training pipeline: initial large-scale Supervised Fine-tuning (SFT) for broad knowledge

acquisition, followed by Group Relative Policy Optimization (GRPO) [46] for fine-grained reasoning refinement on a curated subset.

**ShotQA: A Dedicated Dataset for Cinematography Comprehension.** ShotQA stands as the first large-scale dataset meticulously designed to benchmark and enhance VLMs' grasp of cinematographic techniques. It comprises 58k images and 1.2k video clips. These resources are sourced from 243 diverse films to ensure broad coverage of cinematic styles. All samples are formatted as multi-choice QA pairs, facilitating structured evaluation and targeted training. Each entry is enriched with metadata, including film title and source clip timestamp, allowing for contextual understanding. Table 9 details the sample distribution, revealing a noteworthy balance across most cinematic dimensions. The scale and specificity of ShotQA provide a critical resource for advancing research in this domain.

**Stage 1: Large-scale Supervised Fine-tuning for Foundational Alignment.** In the foundational first stage, ShotVL undergoes SFT using approximately 70k QA pairs sampled from the ShotQA dataset. We utilize Qwen-2.5-VL-3B-Instruct [3] as the base model. The model processes an image or video alongside a question and multiple-choice options, and is trained to directly predict the correct answer via a cross-entropy loss. This SFT phase is crucial for establishing a strong alignment between visual features and specific cinematic terminology, equipping the model with a broad understanding of cinematographic concepts.

**Stage 2: Reinforcement Learning with GRPO for Enhanced Reasoning.** Building upon the SFT-initialized model, the second stage employs GRPO to further elevate ShotVL's reasoning capabilities and prediction accuracy.

Given a multimodal input $x$ (an image/video and textual query), GRPO generates $G$ distinct responses $\{o_1, \ldots, o_G\}$ from the current policy $\pi_{\theta_{\text{old}}}$. These are evaluated using a rule-based binary reward function, inspired by prior work [20, 32, 51]:

$$r(o, x) = \begin{cases} 1, & \text{if } o \text{ is correct (matches the ground truth),} \\ 0, & \text{otherwise.} \end{cases} \tag{1}$$

Following DeepSeek-R1 [20], our reward incorporates two components: (1) a format reward to ensure outputs adhere to a structured pattern (`<think>...</think>` and `<answer>...</answer>` tags), and (2) an accuracy reward comparing the extracted answer from the `<answer>` block with the ground truth.

The advantage $A_i$ for the $i$-th response is calculated by normalizing its reward within the group:

$$A_i = \frac{r_i - \text{mean}(\{r_1, \ldots, r_G\})}{\text{std}(\{r_1, \ldots, r_G\}) + \delta} \tag{2}$$

(where $\delta$ is a small constant for numerical stability, e.g., $1e-8$).

Finally, GRPO optimizes the policy $\pi_\theta$ by maximizing the objective:

$$\mathcal{L}_{\text{GRPO}} = \frac{1}{G} \sum_{i=1}^{G} \min \left( \frac{\pi_\theta(o_i|x)}{\pi_{\theta_{\text{old}}}(o_i|x)} A_i, \text{clip} \left( \frac{\pi_\theta(o_i|x)}{\pi_{\theta_{\text{old}}}(o_i|x)}, 1-\epsilon, 1+\epsilon \right) A_i \right) \tag{3}$$

Here, $\epsilon$ is a hyperparameter controlling the policy update step size, and the clipping mechanism stabilizes training. For this RL phase, we utilize a focused subset of approximately 8k high-quality multiple-choice QA instances from ShotQA to refine the model's ability to select the correct option with higher confidence and precision.

## 5 Experiments

### 5.1 Implementation Details

Our implementation is based on ms-swift [64]. We initialize Qwen2.5-VL-3B-Instruct [3] as our base model. We use around 60k samples for SFT and approximately 8k samples for GRPO. We use Flash Attention-2 [10] as the model's attention implementation and bfloat16 precision for both training and inference to reduce memory consumption. In SFT stage, the global batch size is set to 4, and the model is trained for 1 epoch with a learning rate of 1e-5. In GRPO stage, we set the group size $G$ to 12 and the global batch size to 24. The clipping parameter $\epsilon$ is set to 0.2. The model is trained for 10 epochs with a learning rate of $1 \times 10^{-6}$. Detailed hyper-parameters are provided in the Appendix B.

## 5.2 Main Results

Table 3: Quantitative comparison of GPT-4o [38], Qwen2.5-VL-72B-Instruct [3], and ShotVL (3B, 7B) on ShotBench. Underline indicates previous SOTA in each group.

| Models | SS | SF | CA | LS | LT | LC | SC | CM | Avg |
|---|---|---|---|---|---|---|---|---|---|
| Qwen2.5-VL-72B-Instruct [3] | 75.1 | 82.9 | 56.7 | 46.8 | 59.0 | 49.4 | 54.1 | 48.9 | 59.1 |
| GPT-4o [38] | 69.3 | 83.1 | 58.2 | 48.9 | 63.2 | 48.0 | 55.2 | 48.3 | 59.3 |
| **ShotVL (3B)** | 77.9 | 85.6 | 68.8 | 59.3 | 65.7 | 53.1 | 57.4 | 51.7 | 65.1 |
| **ShotVL (7B)** | 82.5 | 88.8 | 74.1 | 63.8 | 68.1 | 58.6 | 62.6 | 60.6 | 70.1 |

For comparison, we include results from the strongest open-source model (Qwen2.5-VL-72B-Instruct) and the leading proprietary model (GPT-4o) from Table 2, alongside the baseline Qwen2.5-VL-3B-Instruct, as reported in Table 3. Compared to the baseline Qwen2.5-VL-3B-Instruct, Shot-VL (3B) achieves substantial improvements across all dimensions, with an average gain of **19.0** points, demonstrating the effectiveness of our dataset and training methodology. Furthermore, despite having only 3B parameters, our model surpasses both GPT-4o and the strongest open-source model, Qwen2.5-VL-72B-Instruct, setting a **new state of the art** in cinematography language understanding while offering significantly lower deployment and usage costs. We further conduct experiments on the 7B variant of our model and observed even stronger performance, which reinforces the robustness of our dataset and training strategy. We present further experiments and analysis in the following section, with visualizations of representative model outputs included in the Appendix A.2.

## 5.3 Ablation Study

In this section, we investigate the effectiveness of ShotVL's two-stage training strategy. In particular, we compare five training strategies: SFT, CoT-SFT, GRPO, SFT→GRPO, and CoT-SFT→GRPO. For fast exploration, we sample approximately 4k images for the SFT stage and around 1k for GRPO. Besides, we reduce the batch size for SFT to 2, and set the group size and batch size for GRPO to 6. The number of training epochs for GRPO is also reduced to 5, while all other settings remain unchanged. To generate reasoning process for CoT-SFT, we first construct a JSON-formatted knowledge base containing definitions and identification methods for all cinematic terms covered in ShotBench. For each training sample, we retrieve the relevant entries corresponding to the question and candidate choices from the knowledge base, and prompt Gemini-2.0-flash to produce reasoning process grounded in cinematic knowledge. More details of the knowledge base are provided in the Appendix B.

Table 4: Performance comparison of different training strategies. **Bold** indicates the best result, and underline indicates the second best in each group.

| Method | SFT | CoT | GRPO | SS | SF | CA | LS | LT | LC | SC | CM | Avg |
|---|---|---|---|---|---|---|---|---|---|---|---|---|---|
| | | | | 54.6 | 56.6 | 43.1 | 36.6 | 59.3 | 45.1 | 41.5 | 25.8 | 45.3 |
| | ✔ | | | 68.2 | 78.6 | 53.6 | 47.2 | **63.2** | 44.9 | 53.0 | 25.8 | 54.3 |
| | ✔ | ✔ | | 52.6 | 64.3 | 47.3 | 36.4 | 54.8 | 38.0 | 42.2 | 27.4 | 45.4 |
| Choices | | | ✔ | **69.3** | 75.5 | 52.1 | 46.0 | 63.0 | **47.4** | 48.2 | 26.2 | 53.5 |
| | ✔ | ✔ | ✔ | 66.8 | 78.2 | 52.1 | 46.4 | 60.0 | 44.9 | 51.4 | **30.4** | 53.8 |
| | ✔ | | ✔ | 69.1 | **79.3** | **56.7** | **51.1** | 60.5 | 45.4 | 53.2 | 28.6 | **55.5** |

We report the performance of each training method in Table 4. It is observed that all training strategies yield notable improvements over the baseline, demonstrating the high quality and effectiveness of our constructed dataset. Comparing SFT with CoT-SFT, we find that the latter yields very small gains. This may be due to the low quality of reasoning chains generated by Gemini-2.0-flash, which fail to provide effective supervision and may introduce noise. This further highlights the advantage of GRPO, which focuses solely on outcome reward supervision.

Another observation is that reasoning-augmented training consistently improves performance in the camera movement dimension (ranging from +0.4% to +4.6%), despite the ablation experiments being conducted solely on static images and containing no camera movement related questions. This may indicate that reasoning chain generation may implicitly enhance VLMs' capability to recognize dynamic motion. From Figure 7, GRPO consistently improves performance across most dimensions

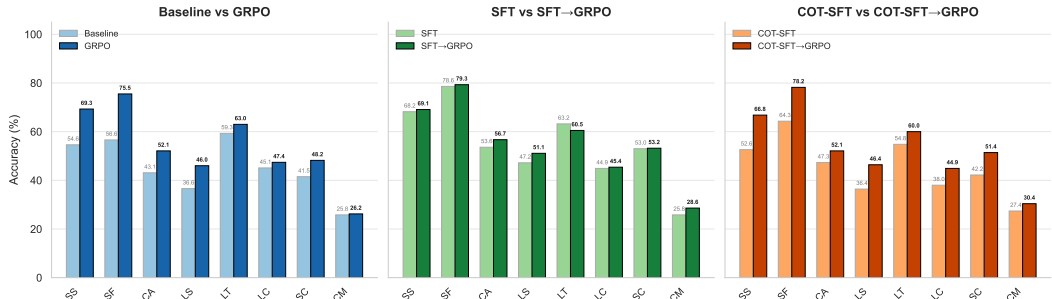

Figure 7: Performance comparison across dimensions before and after applying GRPO under three training setups: baseline, SFT, and CoT-SFT.

under all training settings. Among all configurations, the SFT→GRPO setup achieves the best overall performance, confirming its effectiveness for enhancing cinematography understanding. More case studies are provided in Appendix A.2.

## 6 Conclusion

In this work, we introduce ShotBench, the first comprehensive benchmark designed to evaluate VLMs on cinematography understanding. Through extensive evaluations, we identify notable limitations in current VLMs' capabilities. To tackle these problems, we construct ShotQA, the first large-scale dataset dedicated to this area. We propose ShotVL series with SFT and GRPO training, successfully enhancing the model's (Qwen2.5-VL-3B-Instruct and Qwen2.5-VL-7B-Instruct) capability and achieving new state-of-the-art performance. We hope our work will contribute to future progress in image/video understanding and generation.

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

## Acknowledgments and Disclosure of Funding

**Acknowledgments.** The authors would like to thank the anonymous reviewers for their valuable comments.

**Disclosure of Funding.** This work was funded in part by the National Key R&D Program of China (Grant No. 2023YFC3806000). It was also funded in part by the Ministry of Education, Singapore, under its MOE AcRF Tier 2 (MOE-T2EP20221-0012; MOE-T2EP20223-0002), and by cash and in-kind funding from NTU S-Lab and industry partner(s). It was also funded in part by the MTR Research Funding (MRF) Scheme (CHU-24003), and Shanghai Artificial Intelligence Laboratory.

## A  Discussions

### A.1  Limitations

(1) Both ShotBench and ShotQA are constructed from real-world movie data. However, cinematic shots are not always with standard and clearly defined terminology. Besides, the data distribution is imbalanced across some dimensions (e.g., camera movements like `dolly zoom` is rare in real data). Moreover, high-quality video annotation is labor-intensive. To improve scalability, future work may explore synthetic data for more robust performance.

(2) We primarily validate the effectiveness of our dataset and training approach using Qwen2.5-VL-3B-Instruct, which has limited capability due to a relatively fewer parameters. Further studies may focus on larger base model to further improve performance.

### A.2  ShotVL: Reasoning Like a Cinematographer

**ShotVL Beats GPT-4o**  Our experimental results show that our model outperforms previous SOTA GPT-4o, we visualize some cases and compare the outputs between ShotVL and GPT-4o in Figure 8.

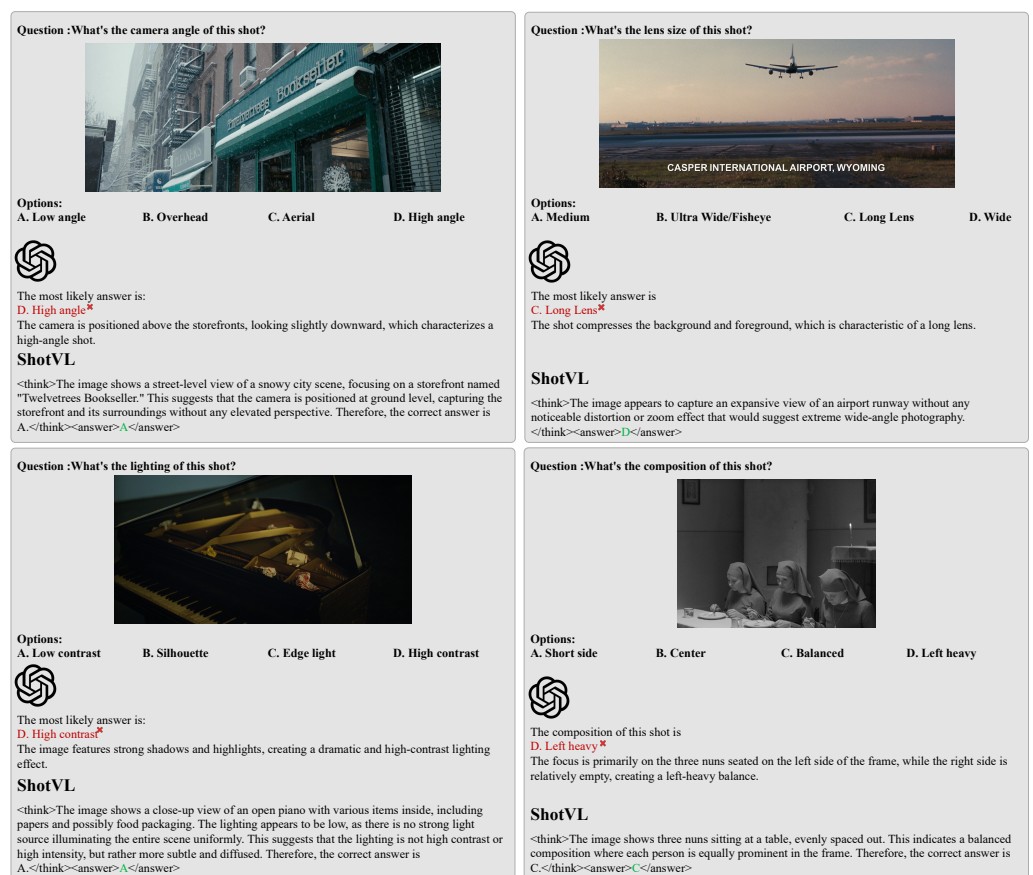

Figure 8: Comparison between GPT-4o and ShotVL.

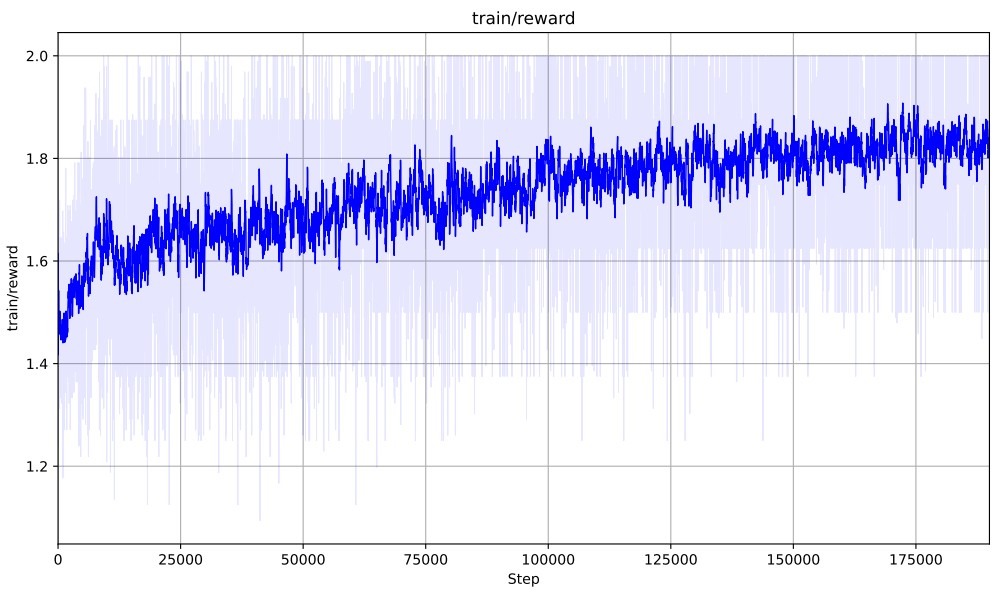

Figure 9: Progression of average reward during GRPO training.

**Reasoning process improves performance**  As illustrated in Figure 9, the average reward increases throughout the GRPO process, indicating that the generation of resoning process helps ShotVL to better understand and recognize cinematic language within a movie shot. More intuitively, we visualize the outputs between ShotVL and the pure SFT variant in Figure 10.

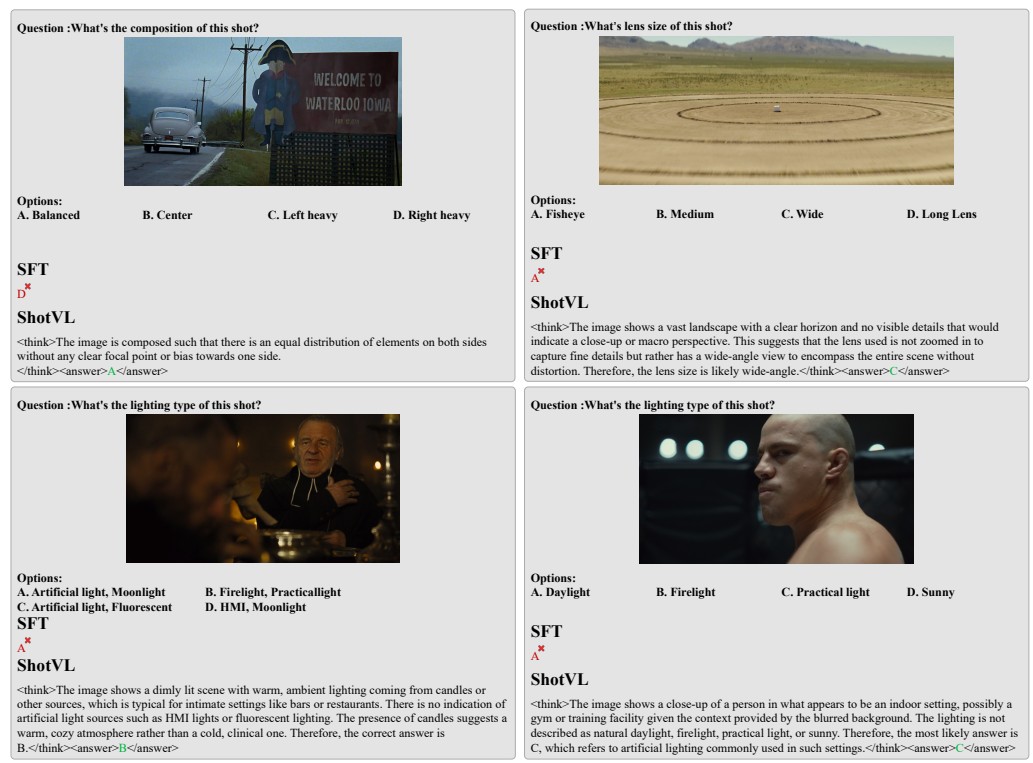

Figure 10: Comparison between GPT-4o and ShotVL.

## A.3  Current VLMs' Cinematography Understanding Needs Further Enhancement

We further visualize failure cases from the strongest open-source model, **Qwen2.5-VL-72B-Instruct**, as well as the strongest proprietary model, **GPT-4o**, as shown in Figure 11.

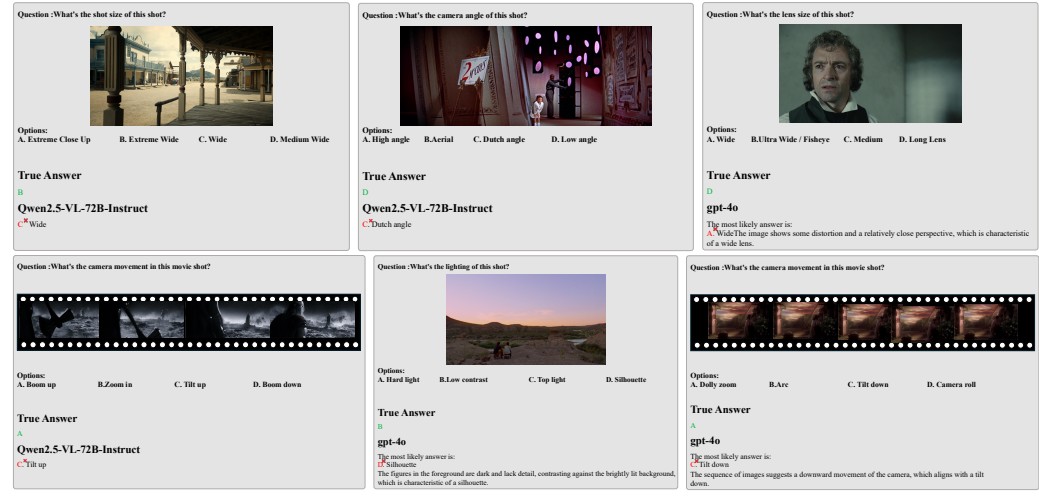

Figure 11: Visualization of failed cases.

## A.4 More Visualizations

More output cases with thinking process from ShotVL are provided in Figure 12.

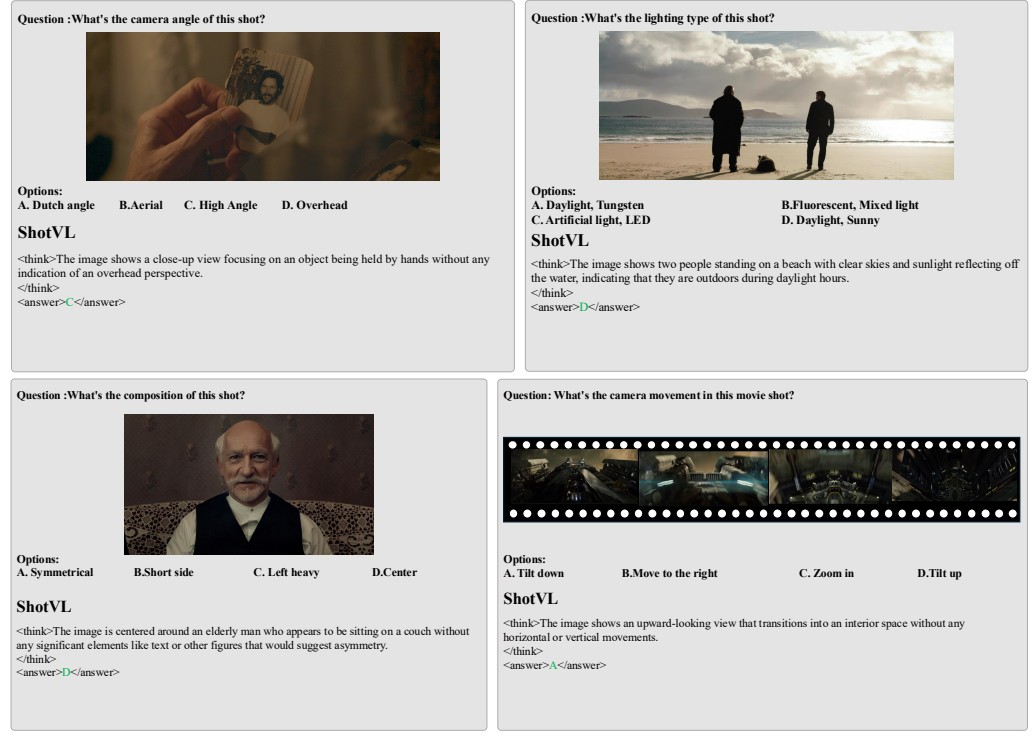

Figure 12: Output visualization of ShotVL.

## B  Implementation Details for Evaluation and Experiments

**Details of Evaluation**    During evaluation, we first attempt to extract each model's final answer using template-based matching. If no valid match is found, we follow the previous works [31, 63] to use GPT-4o as an automatic answer extractor. For open-source models, we densely sample video frames at 12 FPS with a maximum resolution of 360×640 pixels. For image-based samples, we follow the default input configurations of each model. We apply greedy decoding during inference for reproducibility. For proprietary models, we evaluate them via their official APIs, setting the temperature to 0 to produce deterministic outputs. We use GPT-4o (2024-08-06) to extract final answers, based on the prompt in Figure 13.

**Details of Experiments**    Our training implementation is based on ms-swift framework [64]. All hyper-parameters we use for the main experiments are reported in Table 5 and Table 6. The training process of SFT is performed on 4 Nvidia A100 GPUs and the GRPO process is performed on 8 Nvidia A100 GPUs.

In our ablation study, we construct a JSON formatted knowledge base on cinematography and use it to prompt Gemini-2.0-flash to generate reasoning process. We visualize some examples in Figure 14.

## C  Dataset Statistics for ShotBench and ShotQA

**ShotBench**    Below is the list of titles used in constructing the benchmark.

| # | Title | Year | IMDb ID |
|---|-------|------|---------|
| 1 | Manchester by the Sea | 2016 | tt4034228 |

| # | Title | Year | IMDb ID |
|---|-------|------|---------|
| 2 | The Kids Are All Right | 2010 | tt0842926 |
| 3 | Little Women | 2019 | tt3281548 |
| 4 | Flamin Hot | 2023 | tt8105234 |
| 5 | A Quiet Place | 2018 | tt6644200 |
| 6 | Belfast | 2021 | tt12789558 |
| 7 | Green Book | 2018 | tt6966692 |
| 8 | Phantom Thread | 2017 | tt5776858 |
| 9 | Bridge of Spies | 2015 | tt3682448 |
| 10 | 20th Century Women | 2016 | tt4385888 |
| 11 | Passengers | 2016 | tt1355644 |
| 12 | The Fabelmans | 2022 | tt14208870 |
| 13 | Moonlight | 2016 | tt4975722 |
| 14 | Licorice Pizza | 2021 | tt11271038 |
| 15 | BARDO, False Chronicle of a Handful of Truths | 2022 | tt14176542 |
| 16 | Women Talking | 2022 | tt13669038 |
| 17 | Foxcatcher | 2014 | tt1100089 |
| 18 | Christopher Robin | 2018 | tt4575576 |
| 19 | The Great Gatsby | 2013 | tt1343092 |
| 20 | Blade Runner 2049 | 2017 | tt1856101 |
| 21 | Marriage Story | 2019 | tt7653254 |
| 22 | Tinker Tailor Soldier Spy | 2011 | tt1340800 |
| 23 | Nebraska | 2013 | tt1821549 |
| 24 | Black Swan | 2010 | tt0947798 |
| 25 | Youth | 2015 | tt3312830 |
| 26 | The Batman | 2022 | tt1877830 |
| 27 | Mad Max: Fury Road | 2015 | tt1392190 |
| 28 | Minari | 2020 | tt10633456 |
| 29 | Sicario | 2015 | tt3397884 |
| 30 | Knives Out | 2019 | tt8946378 |
| 31 | Ma Raineys Black Bottom | 2020 | tt10514222 |
| 32 | Amour | 2012 | tt1602620 |
| 33 | Lincoln | 2012 | tt0443272 |
| 34 | Judas and the Black Messiah | 2021 | tt9784798 |
| 35 | Life of Pi | 2012 | tt0454876 |
| 36 | Jojo Rabbit | 2019 | tt2584384 |
| 37 | Inside Llewyn Davis | 2013 | tt2042568 |
| 38 | The Banshees of Inisherin | 2022 | tt11813216 |
| 39 | Barbie | 2023 | tt1517268 |
| 40 | The Favourite | 2018 | tt5083738 |
| 41 | Whiplash | 2014 | tt2582802 |
| 42 | Straight Outta Compton | 2015 | tt1398426 |
| 43 | The Revenant | 2015 | tt1663202 |
| 44 | Top Gun: Maverick | 2022 | tt1745960 |
| 45 | Nomadland | 2020 | tt9770150 |
| 46 | Carol | 2015 | tt2402927 |
| 47 | Ad Astra | 2019 | tt2935510 |
| 48 | RRR | 2022 | tt8178634 |
| 49 | Midnight in Paris | 2011 | tt1605783 |
| 50 | A Separation | 2011 | tt1832382 |
| 51 | The Hobbit: The Desolation of Smaug | 2013 | tt1170358 |
| 52 | The Worst Person in the World | 2021 | tt10370710 |
| 53 | The Power of the Dog | 2021 | tt10293406 |
| 54 | Alice in Wonderland | 2010 | tt1014759 |
| 55 | TÁR | 2022 | tt14444726 |
| 56 | Can You Ever Forgive Me? | 2018 | tt4595882 |
| 57 | Ted | 2012 | tt1637725 |
| 58 | Hugo | 2011 | tt0970179 |
| 59 | Cold War | 2018 | tt6543652 |
| 60 | May December | 2023 | tt13651794 |
| 61 | Her | 2013 | tt1798709 |
| 62 | Unbroken | 2014 | tt1809398 |
| 63 | Ida | 2013 | tt2718492 |
| 64 | Ex Machina | 2014 | tt0470752 |

| # | Title | Year | IMDb ID |
|---|---|---|---|
| 65 | Beyond the Lights | 2014 | tt3125324 |
| 66 | The Kings Speech | 2010 | tt1504320 |
| 67 | American Sniper | 2014 | tt2179136 |
| 68 | The Imitation Game | 2014 | tt2084970 |
| 69 | Before Midnight | 2013 | tt2209418 |
| 70 | Promising Young Woman | 2020 | tt9620292 |
| 71 | Baby Driver | 2017 | tt3890160 |
| 72 | Indiana Jones and the Dial of Destiny | 2023 | tt1462764 |
| 73 | Captain Phillips | 2013 | tt1535109 |
| 74 | Glass Onion: A Knives Out Mystery | 2022 | tt24734444 |
| 75 | The Disaster Artist | 2017 | tt3521126 |
| 76 | Never Look Away | 2018 | tt5311542 |
| 77 | Hail, Caesar! | 2016 | tt0475290 |
| 78 | Star Trek Into Darkness | 2013 | tt1408101 |
| 79 | Nightmare Alley | 2021 | tt7740496 |
| 80 | All Quiet on the Western Front | 2022 | tt1016150 |
| 81 | Fences | 2016 | tt2671706 |
| 82 | Harriet | 2019 | tt4648786 |
| 83 | Zero Dark Thirty | 2012 | tt1790885 |
| 84 | No Time to Die | 2021 | tt2382320 |
| 85 | Get Out | 2017 | tt5052448 |
| 86 | Moneyball | 2011 | tt1210166 |
| 87 | Skyfall | 2012 | tt1074638 |
| 88 | Living | 2022 | tt9051908 |
| 89 | The Lobster | 2015 | tt3464902 |
| 90 | The Big Sick | 2017 | tt5462602 |
| 91 | Spectre | 2015 | tt2379713 |
| 92 | Napoleon | 2023 | tt13287846 |
| 93 | El Conde | 2023 | tt21113540 |
| 94 | Lion | 2016 | tt3741834 |
| 95 | Arrival | 2016 | tt2543164 |
| 96 | Parasite | 2019 | tt6751668 |
| 97 | The Lost Daughter | 2021 | tt9100054 |
| 98 | Gravity | 2013 | tt1454468 |
| 99 | The White Tiger | 2021 | tt6571548 |
| 100 | Mank | 2020 | tt10618286 |
| 101 | The Trial of the Chicago 7 | 2020 | tt1070874 |
| 102 | Maestro | 2023 | tt5535276 |
| 103 | Silence | 2016 | tt0490215 |
| 104 | Drive My Car | 2021 | tt14039582 |
| 105 | Silver Linings Playbook | 2012 | tt1045658 |
| 106 | Logan | 2017 | tt3315342 |
| 107 | The Hobbit: The Battle of the Five Armies | 2014 | tt2310332 |
| 108 | Moonrise Kingdom | 2012 | tt1748122 |
| 109 | Room | 2015 | tt3170832 |
| 110 | Triangle of Sadness | 2022 | tt7322224 |
| 111 | Real Steel | 2011 | tt0433035 |
| 112 | The Post | 2017 | tt6294822 |
| 113 | Roma | 2018 | tt6155172 |
| 114 | If Beale Street Could Talk | 2018 | tt7125860 |
| 115 | The Ballad of Buster Scruggs | 2018 | tt6412452 |
| 116 | Django Unchained | 2012 | tt1853728 |
| 117 | The Lighthouse | 2019 | tt7984734 |
| 118 | A Star Is Born | 2018 | tt1517451 |
| 119 | The Descendants | 2011 | tt1033575 |
| 120 | Babylon | 2022 | tt10640346 |
| 121 | Once Upon a Time in Hollywood | 2016 | tt4010884 |
| 122 | The Shape of Water | 2017 | tt5580390 |
| 123 | King Richard | 2021 | tt9620288 |
| 124 | Lady Bird | 2017 | tt4925292 |
| 125 | Joker | 2019 | tt7286456 |
| 126 | The Danish Girl | 2015 | tt0810819 |
| 127 | Winters Bone | 2010 | tt1399683 |

| # | Title | Year | IMDb ID |
|---|-------|------|---------|
| 128 | La La Land | 2016 | tt3783958 |
| 129 | Beasts of the Southern Wild | 2012 | tt2125435 |
| 130 | Da 5 Bloods | 2020 | tt9777644 |
| 131 | The Irishman | 2019 | tt1302006 |
| 132 | Darkest Hour | 2017 | tt4555426 |
| 133 | The Father | 2020 | tt10272386 |
| 134 | BlacKkKlansman | 2018 | tt7349662 |
| 135 | 12 Years a Slave | 2013 | tt2024544 |
| 136 | Adaptation. | 2002 | tt0268126 |
| 137 | Bridesmaids | 2011 | tt1478338 |
| 138 | Vice | 2018 | tt6266538 |
| 139 | The Girl with the Dragon Tattoo | 2011 | tt1568346 |
| 140 | The Hobbit: An Unexpected Journey | 2012 | tt0903624 |
| 141 | Into the Woods | 2014 | tt2180411 |
| 142 | Boyhood | 2014 | tt1065073 |
| 143 | First Man | 2018 | tt1213641 |
| 144 | Parallel Mothers | 2021 | tt12618926 |
| 145 | The Martian | 2015 | tt3659388 |
| 146 | The Social Network | 2010 | tt1285016 |
| 147 | First Reformed | 2017 | tt6053438 |
| 148 | Deepwater Horizon | 2016 | tt1860357 |
| 149 | The Wolf of Wall Street | 2013 | tt0993846 |
| 150 | Dallas Buyers Club | 2013 | tt0790636 |
| 151 | Hacksaw Ridge | 2016 | tt2119532 |
| 152 | Dunkirk | 2017 | tt5013056 |
| 153 | Selma | 2014 | tt1020072 |
| 154 | The Theory of Everything | 2014 | tt2980516 |
| 155 | Nightcrawler | 2014 | tt2872718 |
| 156 | Everything Everywhere All at Once | 2022 | tt6710474 |
| 157 | True Grit | 2010 | tt1403865 |
| 158 | Jackie | 2016 | tt1619029 |
| 159 | Killers of the Flower Moon | 2023 | tt5537002 |
| 160 | One Night in Miami... | 2020 | tt10612922 |
| 161 | Ford v Ferrari | 2019 | tt1950186 |
| 162 | Anatomy of a Fall | 2023 | tt17009710 |
| 163 | The Midnight Sky | 2020 | tt10539608 |
| 164 | The Tree of Life | 2011 | tt0478304 |
| 165 | Brooklyn | 2015 | tt2381111 |
| 166 | American Hustle | 2013 | tt1800241 |
| 167 | The Lone Ranger | 2013 | tt1210819 |
| 168 | Mudbound | 2017 | tt2396589 |
| 169 | Oppenheimer | 2023 | tt15398776 |
| 170 | Another Round | 2020 | tt10288566 |
| 171 | Fifty Shades of Grey | 2015 | tt2322441 |
| 172 | Argo | 2012 | tt1024648 |
| 173 | The Two Popes | 2019 | tt8404614 |
| 174 | Elvis | 2022 | tt3704428 |
| 175 | Hell or High Water | 2016 | tt2582782 |
| 176 | The Hateful Eight | 2015 | tt3460252 |
| 177 | Molly's Game | 2017 | tt4209788 |
| 178 | News of the World | 2020 | tt6878306 |
| 179 | The Adventures of Tintin | 2011 | tt0983193 |
| 180 | The Greatest Showman | 2017 | tt1485796 |
| 181 | Empire of Light | 2022 | tt14402146 |
| 182 | Interstellar | 2014 | tt0816692 |
| 183 | Extremely Loud & Incredibly Close | 2011 | tt0477302 |
| 184 | 1917 | 2019 | tt8579674 |
| 185 | 127 Hours | 2010 | tt1542344 |
| 186 | Spotlight | 2015 | tt1895587 |
| 187 | Rocketman | 2019 | tt2066051 |
| 188 | Marshall | 2017 | tt5301662 |
| 189 | Drive | 2011 | tt0780504 |
| 190 | Inherent Vice | 2014 | tt1791528 |

| #   | Title                                            | Year | IMDb ID     |
|-----|--------------------------------------------------|------|-------------|
| 191 | Tangled                                          | 2010 | tt0398286   |
| 192 | Captain America: The Winter Soldier              | 2014 | tt1843866   |
| 193 | The Big Short                                    | 2015 | tt1596363   |
| 194 | Tenet                                            | 2020 | tt6723592   |
| 195 | Sound of Metal                                   | 2019 | tt5363618   |
| 196 | The Holdovers                                    | 2023 | tt14849194  |
| 197 | Les Misérables                                   | 2012 | tt1707386   |
| 198 | Call Me by Your Name                             | 2017 | tt5726616   |
| 199 | Hidden Figures                                   | 2016 | tt4846340   |
| 200 | The Grand Budapest Hotel                         | 2014 | tt2278388   |
| 201 | Past Lives                                        | 2023 | tt13238346  |
| 202 | Dont Look Up                                      | 2018 | tt6134232   |
| 203 | Poor Things                                       | 2023 | tt14230458  |
| 204 | Mr. Turner                                        | 2014 | tt2473794   |
| 205 | Prisoners                                         | 2013 | tt1392214   |
| 206 | Casino Royale                                     | 2006 | tt0381061   |
| 207 | Polytechnique                                     | 2009 | tt1194238   |
| 208 | Gladiator                                         | 2000 | tt0172495   |
| 209 | 47 Ronin                                          | 2013 | tt1335975   |
| 210 | Mindhunters                                       | 2004 | tt0297284   |
| 211 | The Curious Case of Benjamin Button               | 2008 | tt0421715   |
| 212 | Mission: Impossible                               | 2011 | tt1229238   |
| 213 | Wednesday                                         | 2007 | tt1024676   |
| 214 | No Country for Old Men                            | 2007 | tt0477348   |
| 215 | The Last Duel                                     | 2021 | tt4244994   |
| 216 | Rebel Moon                                        | 2023 | tt14998742  |
| 217 | Thor: Love and Thunder                            | 2022 | tt10648342  |
| 218 | One Piece                                         | 2018 | tt10109772  |
| 219 | Star Wars                                         | 1977 | tt0076759   |
| 220 | The Witcher                                       | 2017 | tt7351402   |
| 221 | The Lord of the Rings: The Rings of Power         | 2022 | tt7631058   |
| 222 | There Will Be Blood                               | 2007 | tt0469494   |
| 223 | Bullet Train                                      | 2022 | tt12593682  |
| 224 | Quantum of Solace                                 | 2008 | tt0830515   |
| 225 | Dune: Part Two                                    | 2024 | tt15239678  |
| 226 | Forrest Gump                                      | 1994 | tt0109830   |
| 227 | Loki season 2                                     | 2023 | tt18271346  |
| 228 | Cruella                                           | 2021 | tt3228774   |
| 229 | Fight Club                                        | 1999 | tt0137523   |
| 230 | The Fall Guy                                      | 2024 | tt1684562   |
| 231 | James Bond                                        | 2015 | tt4896340   |
| 232 | Mission: Impossible                               | 2011 | tt1229238   |
| 233 | Scott Pilgrim vs. the World                       | 2010 | tt0446029   |
| 234 | John Wick                                         | 2014 | tt2911666   |
| 235 | The Suicide Squad                                 | 2021 | tt6334354   |
| 236 | The Killer                                        | 2023 | tt1136617   |
| 237 | Superman                                          | 1978 | tt0078346   |
| 238 | Inception                                         | 2010 | tt1375666   |
| 239 | World of Warcraft                                 | 2014 | tt4191810   |
| 240 | The Raid                                          | 1954 | tt0047388   |
| 241 | Barry Lyndon                                      | 1975 | tt0072684   |
| 242 | Captain America: Civil War                        | 2016 | tt3498820   |
| 243 | Marvels Jessica Jones                             | 2015 | tt2357547   |

An overview of cinematic terms used in ShotBench and the distribution of QA pairs are visualized in Figure 15a and Figure 15b. Details of ShotBench are provided in Table 8. We adopted an aesthetic score threshold of 3.0, samples with low aesthetic scores often characterized by poor composition, motion blur, or chaotic visual content. Such samples typically lack well-defined cinematic attributes and were excluded to maintain the overall quality of our dataset. And a total of 20 trained annotators participated in the annotation process during dataset construction.

Figure 13: Prompt format used for answer extraction from GPT-4o.

Table 5: Hyper-parameters for SFT.

| Parameter | Value |
| --- | --- |
| model | Qwen2.5-VL-3B-Instruct |
| attn_impl | flash_attn |
| train_type | full |
| torch_dtype | bfloat16 |
| num_train_epochs | 1 |
| per_device_train_batch_size | 1 |
| per_device_eval_batch_size | 1 |
| learning_rate | 1e-5 |
| gradient_accumulation_steps | 16 |
| eval_steps | 100 |
| save_steps | 100 |
| save_total_limit | 3 |
| logging_steps | 5 |
| max_length | 3072 |
| system | "You are a helpful assistant." |
| warmup_ratio | 0.05 |
| dataloader_num_workers | 4 |

Table 6: Hyper-parameters for GRPO.

| Parameter | Value |
| --- | --- |
| model | Qwen2.5-VL-3B After SFT |
| rlhf_type | grpo |
| use_vllm | true |
| vllm_device | auto |
| vllm_gpu_memory_utilization | 0.6 |
| train_type | full |
| torch_dtype | bfloat16 |
| max_length | 2048 |
| max_completion_length | 1024 |
| num_train_epochs | 10 |
| per_device_train_batch_size | 4 |
| per_device_eval_batch_size | 4 |
| learning_rate | 1e-6 |
| gradient_accumulation_steps | 4 |
| save_strategy | steps |
| eval_strategy | steps |
| eval_steps | 500 |
| save_steps | 500 |
| save_total_limit | 3 |
| logging_steps | 1 |
| warmup_ratio | 0.01 |
| dataloader_num_workers | 12 |
| num_generations | 12 |
| temperature | 1.0 |
| repetition_penalty | 1.1 |
| deepspeed | zero3 |
| num_iterations | 1 |
| num_infer_workers | 2 |
| async_generate | false |
| beta | 0.001 |
| max_grad_norm | 0.5 |

Table 8: Distribution of cinematic terms used in ShotBench

| Dimension | Term | % |
|---|---|---|
| Shot Size | Wide | 13.3 |
| | Close Up | 13.1 |
| | Extreme Wide | 13.1 |
| | Medium Close Up | 12.6 |
| | Medium Wide | 12.1 |
| | Medium | 11.8 |
| | Extreme Close Up | 11.6 |
| Shot Framing | Single | 15.4 |
| | Insert | 14.8 |
| | 2 shot | 14.5 |
| | Group shot | 14.2 |
| | Establishing shot | 14.2 |
| | Over the shoulder | 13.6 |
| | 3 shot | 13.5 |
| Camera Angle | Aerial | 20.7 |
| | Overhead | 20.1 |
| | Low angle | 19.8 |
| | High angle | 19.7 |
| | Dutch angle | 19.7 |
| Lens Size | Long Lens | 25.5 |
| | Wide | 25.2 |
| | Ultra Wide&Fisheye | 24.7 |
| | Medium | 24.7 |
| Lighting Type | Daylight | 12.7 |
| | Artificial light | 11.9 |
| | Mixed light | 10.5 |
| | Firelight | 10.0 |
| | Overcast | 9.4 |
| | Practical light | 9.4 |
| | Sunny | 9.3 |
| | Moonlight | 8.8 |
| | Fluorescent | 8.6 |
| | HMI | 7.7 |
| | Tungsten | 1.2 |
| | LED | 0.8 |
| Lighting Condition | Side light | 10.8 |
| | Backlight | 10.7 |
| | High contrast | 10.3 |
| | Silhouette | 10.2 |
| | Edge light | 10.1 |
| | Underlight | 10.1 |
| | Top light | 10.0 |
| | Hard light | 10.0 |
| | Soft light | 9.8 |
| | Low contrast | 8.2 |
| Composition | Center | 17.4 |
| | Balanced | 17.1 |
| | Symmetrical | 16.7 |
| | Right heavy | 16.4 |
| | Left heavy | 16.3 |
| | Short side | 16.2 |
| Camera Movement | Push in | 10.4 |
| | Pull out | 9.1 |
| | Boom up | 8.5 |
| | Pan left | 7.7 |
| | Pan right | 7.4 |
| | Tilt down | 7.1 |
| | Tilt up | 6.8 |
| | Boom down | 6.5 |
| | Zoom in | 6.4 |
| | Static | 6.3 |
| | Move to the right | 5.9 |
| | Move to the left | 4.7 |
| | Zoom out | 4.5 |
| | Arc | 3.9 |
| | Camera roll | 3.7 |
| | Dolly zoom | 0.6 |

…
{
"name": "Wide Shot / Long Shot",
"abbreviation": "WS / LS",
"meaning": "Shows the subject from head to toe, although not necessarily filling the frame. The focus is still largely on the environment, but the subject is more prominent than in an EWS. It shows the subject within their surroundings, providing context.",
"identification": "Identify the full body of the subject within the frame, usually with considerable space above their head and below their feet. The setting is clearly visible and important, but the character is identifiable and their full figure is shown."
},
{
"name": "Close Up",
"abbreviation": "CU",
"meaning": "Frames a specific part of the subject, typically the head and face, from the neck up. It tightly frames the character's face, emphasizing their emotional state and reactions. Little to no background is visible.",
"identification": "The subject's head/face fills most of the frame. The top of the frame is usually just above the hairline, and the bottom is below the chin, often showing the neck or very top of the shoulders. Facial details and emotions are the primary focus."
},
{
"name": "Extreme Close Up",
"abbreviation": "ECU",
"meaning": "Frames only a small portion or detail of the subject, such as the eyes, mouth, or a specific object. It is used to create intensity, highlight a crucial detail, or convey strong emotion or tension.",
"identification": "The shot magnifies a single feature (like eyes, lips) or a small object, filling the entire screen with it. You cannot see the whole face or the broader context. It often feels intimate or intense."
}
…

Figure 14: Examples of constructed knowledge base on cinematic language.

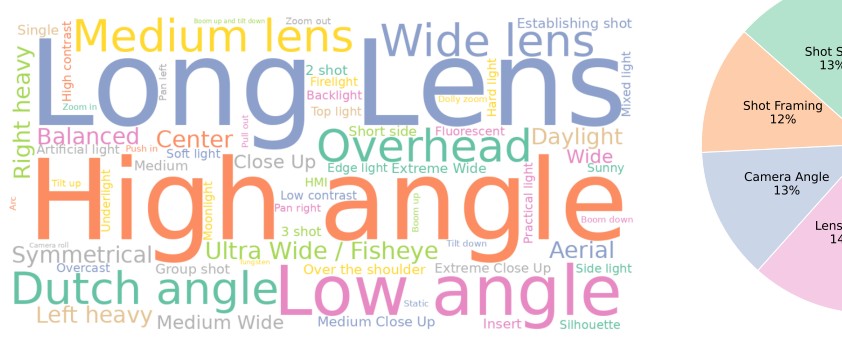

(a) Overview of cinematic terms used in ShotBench.   (b) Distribution of questions.

Figure 15: Statistics of ShotBench across different dimensions.

**More details of ShotQA** ShotQA is constructed in a similar manner to ShotBench (Section 3.2), except that only video samples are totally manually annotated and verified by trained annotators and experts. For large-scale image annotations sourced from expert cinematography websites, we conducted random sampling checks and found their quality adequate for training use. We adopt a two-stage filtering strategy to ensure no overlap between training and evaluation sets: we first remove duplicate samples based on IMDb IDs and timestamp as a coarse-level filtering step. Then, we extract CLIP features for all samples and exclude samples from the training set whose feature has a cosine similarity greater than 0.95 (following [5, 6, 19]) with any sample in ShotBench.

The GRPO sub-dataset consists of a combination of mid-difficulty samples and uniformly sampled QA pairs across all eight dimensions. We identify mid-difficulty samples by prompting Qwen2.5-VL-3B to answer a subset of the data multiple times and selecting those for which both correct and incorrect answers are observed across different runs. The remaining QA pairs were uniformly sampled again and used for SFT.

Table 9: Sample distribution in the ShotQA.

| Dimension | #Samples |
|---|---|
| Camera Angle (CA) | 9,405 |
| Shot Composition (SC) | 9,597 |
| Lens Size (LS) | 8,324 |
| Lighting Condition (LC) | 8,778 |
| Lighting Type (LT) | 6,811 |
| Shot Framing (SF) | 8,298 |
| Shot Size (SS) | 8,579 |
| Camera Movement (CM) | 1,200 |

## D    Reference Materials on Cinematography Used in ShotBench Construction.

During the construction of ShotBench, we trained annotators through professional teaching websites and teaching videos publicly available on the Internet. We provide some representative materials in Table 10

Table 10: Representative reference materials used to train annotators.

| Dimension | Website | Video |
|---|---|---|
| Shot Size | https://www.studiobinder.com/blog/types-of-camera-shots-sizes-in-film/ | https://www.youtube.com/watch?v=AyML8xuKfoc |
| Shot Framing | https://www.studiobinder.com/blog/types-of-camera-shot-frames-in-film/ | https://www.youtube.com/watch?v=qQNiqzuXjoM |
| Camera Angle | https://www.studiobinder.com/blog/types-of-camera-shot-angles-in-film/ | https://www.youtube.com/watch?v=wLfZL9PZI9k |
| Lens Size | https://www.studiobinder.com/blog/focal-length-camera-lenses-explained/ | https://www.youtube.com/watch?v=uSsIqR3DuK8 |
| Lighting | https://www.studiobinder.com/blog/film-lighting/ | https://www.youtube.com/watch?v=r2nD_knsNrc |
| Shot Composition | https://www.studiobinder.com/blog/rules-of-shot-composition-in-film/ | https://www.youtube.com/watch?v=hUmZ1dtODTg&t=10s |
| Camera Movement | https://www.studiobinder.com/blog/different-types-of-camera-movements-in-film/ | https://www.youtube.com/watch?v=IiyBo-qLDeM |

## E    Societal Impact Statement

This work presents ShotBench, a benchmark for evaluating vision-language models (VLMs) on cinematic language understanding, and ShotQA, a large-scale dataset designed for training such capabilities. Additionally, we propose ShotVL, a reasoning-enhanced VLM series trained via SFT and GRPO.

**Positive Societal Impact.** By improving VLMs' understanding of professional cinematic conventions, our work can contribute to the development of AI systems that assist in film production. Specifically, cinematography-aware models may support AI-assisted filmmaking tasks such as shot planning, automated style matching, and film-level image/video generation. These capabilities could help democratize access to professional filmmaking workflows, reduce production costs, and empower creators with limited resources. In addition, our benchmark and dataset may foster research into multimodal reasoning, benefiting broader applications in video understanding and generation.

**Negative Societal Impact.** As with other generative or vision-language technologies, there are potential negative applications. For example:

Disinformation and deepfakes: Enhanced understanding of cinematic language could be exploited to make AI-generated fake content more visually convincing or emotionally manipulative.

Creative job displacement: The use of cinematography-aware models in automated filmmaking pipelines may marginalize certain creative roles (e.g., assistant editors, junior cinematographers).

Bias propagation: If the training data or annotations reflect specific cultural aesthetics or norms (e.g., Western cinematic styles), the resulting models may encode biased visual preferences or overlook underrepresented filmmaking traditions.

