# OpenReview forum: "ShotBench: Expert-Level Cinematic Understanding in Vision-Language Models"
_NeurIPS.cc/2025/Conference — NeurIPS 2025 poster_

### Official Review · Reviewer_xwhm · 2025-06-25

**Clarity:** 2
**Significance:** 3
**Originality:** 4
**Rating:** 5
**Confidence:** 3

**Summary:**

The authors introduce ShotQA, a dataset for cinematographic understanding, along with ShotBench, the associated benchmark. They also propose the ShotVL model, trained on the ShotQA dataset and evaluated using the ShotBench benchmark. Their proposed model achieves substantial improvements over existing approaches.

**Questions:**

* Question 1: Was the same video clip potentially annotated by more than one annotator? If so, did the authors compute inter-annotator agreement (IAA) to assess consistency between annotators?

* Question 2: Is it possible that clips from the same movie appear in both the training and test sets? If so, do the authors consider this a potential issue?

* Question 3: Line 125-126: What does "low-quality sample" mean in this context? How can two clips from the same film differ in quality?

**Ethical Concerns:**

["NO or VERY MINOR ethics concerns only"]

**Final Justification:**

Having considered the authors’ replies to my questions, along with the other reviewers’ comments, I am choosing to maintain my initial evaluation.

**Limitations:**

Yes

**Quality:**

3

**Strengths And Weaknesses:**

**Strengths:**  This is the first dataset and benchmark designed to evaluate vision-language models (VLMs) on their ability to understand cinematic language.The annotations have been validated by experts, ensuring their reliability. The authors establish comprehensive baselines using 24 VLMs, including both open-source and proprietary models. Additionally, an ablation study supports the effectiveness of their ShotVL model training strategy.

**Weaknessess:**
* Minor comment: reference 17 and 18 correspond to the same paper
* There is a lack of detail regarding the annotation process (see Question 1 and 2) as well as the data separation process (Question 3)

---

> ### Author Rebuttal · Authors · 2025-07-30
>
> Thank you for the support and for recognizing the novelty and impact of our work. We have carefully reviewed your questions and provide point-by-point responses below:
>
> ### **Q1: Was Inter-Annotator Agreement (IAA) computed?**
>
> While we did not compute a formal inter-annotator agreement score, we employed a **rigorous multi-stage annotation and verification protocol to ensure quality and consistency** (as described in Section 3.2). All annotations were evenly divided into batches, each of which was audited by experts. A batch was only accepted if it achieved an accuracy above 90%; otherwise, it was sent back for re-annotation. Whenever a batch was rejected, annotators and experts engaged in discussion to reach a consensus. **We believe this expert-in-the-loop approach offers stronger guarantees of annotation quality than relying solely on IAA scores**.
>
> ### **Q2: Is it possible that clips from the same movie appear in both training and test sets?**
>
> We confirm that **there are no duplicate samples between the training and test sets.** To ensure this, we compute CLIP features for all samples and remove any training sample with cosine similarity greater than 0.95 (following [1,2,3]) to a test sample.
>
> To clarify, our benchmark aims to evaluate fine-grained cinematic understanding, which often requires distinguishing between **visually similar but cinematographically distinct shots** (e.g., different shot sizes or compositions from the same scene). Therefore, we **allow different shots from the same movie to appear in both splits**, as this better reflects real-world challenges in cinematic understanding.
>
> We will clarify these points in the revised paper.
>
> ### **Q3: What does "low-quality sample" mean?**
>
> The "low-quality samples" mentioned in Line 125–126 refer to those filtered out using the LAION aesthetic predictor. These are samples with low aesthetic scores, often characterized by **poor composition**, **motion blur**, or **chaotic visual content**. Such samples typically lack well-defined cinematic attributes and were excluded to maintain the overall quality and clarity of our dataset.
>
> ### **Minor comment on references**
>
> Thank you for your careful proofreading. We will merge the duplicate references (17 and 18) accordingly.
>
> ### References
>
> > [1] Gadre S Y, Ilharco G, Fang A, et al. Datacomp: In search of the next generation of multimodal datasets[J]. Advances in Neural Information Processing Systems, 2023, 36: 27092-27112.
> >
> > [2] Caron M, Iscen A, Fathi A, et al. A generative approach for wikipedia-scale visual entity recognition[C]//Proceedings of the IEEE/CVF Conference on Computer Vision and Pattern Recognition. 2024: 17313-17322.
> >
> > [3] Caron M, Fathi A, Schmid C, et al. Web-scale visual entity recognition: An LLM-driven data approach[J]. Advances in Neural Information Processing Systems, 2024, 37: 34533-34560.

---

> ### Author Response · Authors · 2025-08-04
>
> Please let us know if there are any remaining questions or suggestions, we sincerely value your feedback.

---

> > ### Comment · Reviewer_xwhm · 2025-08-06
> >
> > Thank you for addressing my questions. After considering the reviews and the authors’ rebuttal, I have decided to maintain my original rating.

---

### Official Review · Reviewer_SCDV · 2025-06-30

**Clarity:** 2
**Significance:** 3
**Originality:** 3
**Rating:** 4
**Confidence:** 4

**Summary:**

The paper introduces ShotBench, a benchmark designed to evaluate vision-language models (VLMs) on their understanding of cinematic language across eight key dimensions (e.g., shot size, camera angle, composition). It presents ShotBench with 3,608 high-quality QA pairs built from images and clips of over 200 films. The authors also construct ShotQA, a 70k QA dataset, and propose ShotVL, a new VLM trained with supervised fine-tuning (SFT) and Group Relative Policy Optimization (GRPO), achieving performance gains over state-of-the-art models like GPT-4o. Experiments show ShotVL sets a new state-of-the-art in cinematic reasoning, despite its small size (3B parameters).

**Questions:**

1. I have a few questions about the datasets and experimental setup:
- Will all three datasets - ShotBench, the 70k QA pairs for SFT, and the 8k QA pairs for GRPO - be made publicly available?
- While the construction of the 3,608 QA pairs in ShotBench is well explained, the process and attributes of the 70k and 8k QA datasets are not clearly described. Could you provide more details on how these were created, perhaps in the appendix? It would also be helpful to include detailed features and attribute values for all datasets planned for release.
- Could you clarify whether the datasets used for SFT and GRPO training are completely separate from ShotBench, ensuring that evaluation was done on truly unseen QA pairs? Clear information on the training, validation, and test splits would increase confidence in the reported performance improvements of ShotVL.

2. Could you clarify whether permission was obtained to use the movie clips for research purposes?

3. While this study covers eight dimensions comprehensively, did you conduct any comparisons of performance on individual dimensions with existing cinematography understanding methods, such as HunyuanVideo or SGNet?

4. In ShotBench, for questions requiring inference of multiple keywords as answers - such as the example of “Lighting Condition” shown at the bottom left of Figure 1 - are models expected to mention all keywords? It would be helpful for readers if you could clarify how performance was measured and evaluated in such cases.

5. Have you tried applying your approach to larger models, such as Qwen2.5-VL-7B-Instruct or InternVL3-8B? I am curious about the potential effects and performance improvements on these more capable models.

If these questions and concerns are adequately addressed, I would be willing to consider raising my score.

**Ethical Concerns:**

["Major Concern: Data privacy, copyright, and consent"]

**Final Justification:**

Considering the authors’ comprehensive description of the data sources and the additional experiments they conducted, they have fully addressed my concerns. Consequently, I have raised my evaluation score to reflect the substantially strengthened manuscript.

**Limitations:**

Yes, but further clarification is needed. Clearly stating whether the use of movie clips for research purposes meets copyright and licensing requirements could help prevent potential legal issues.

**Paper Formatting Concerns:**

- On Abstract, line 12, VLMs capability → VLMs’ capability
- The caption of Table 2 does not match the content of the table and appears to be a duplicate of Table 3. Please update the caption to accurately reflect the table’s content.
- On page 6, line 180, please provide the full form of the acronym “MCU” in parentheses upon first use for clarity.
- On page 7, line 210, consider removing the phrase “(second row, third column)” as it appears unnecessary and may confuse readers.
- On page 8, line 266, state of the art → state-of-the-art

**Quality:**

2

**Strengths And Weaknesses:**

Strengths
1. Generative models are increasingly applied to image and video creation across various domains, including commercial applications, yet they often fail to produce outputs that accurately align with prompts, particularly in cinematic contexts. This paper makes a contribution by developing a benchmark to test how well vision-language models (VLMs) understand cinematic language, which is essential for overcoming these limitations. The authors also curated a cinematic knowledge QA dataset, which they used to train ShotVL. By focusing on understanding cinematic language - a critical capability for applications in film production, video editing, and content generation - the research demonstrates practical value.
2. The authors first organized expert knowledge in this domain to define eight core dimensions of cinematography and their corresponding attributes, then extended this structured framework to construct ShotBench, ShotQA, and ShotVL. This systematic research and development approach, combined with the commitment to make the resources publicly available, is a major strength of the work, as it will stimulate further research in the field and enhance the applicability and usability of the proposed benchmark and datasets in real-world scenarios.
3. The paper thoroughly analyzes 24 diverse VLMs using standardized metrics, revealing the current limitations of these models.

Weaknesses
1. The paper does not clearly state the sources of the movie clips used or whether copyright permissions were obtained. The authors mention in Section 3.2 (Data Construction Process) that “Data are sourced from public websites and include high-resolution images and video clips,” but since the dataset includes clips from recent and high-profile films, including those nominated for or winning Academy Awards, it would be helpful to clarify whether the authors secured the necessary rights or permissions to use these materials for research. There have been cases where research groups were unable to secure usage rights for copyrighted film content and had to abandon or significantly modify their projects. Given current concerns about intellectual property and ethical use of data, a clear statement on this issue would improve the transparency of the work.
2. The paper introduces three datasets: ShotBench, which includes 3,608 high-quality QA pairs annotated by trained experts; ShotQA, consisting of 70k QA pairs used for SFT training; and an additional 8k high-quality QA pairs used for GRPO training. While the construction process of ShotBench is described in detail, there is very limited information provided on how the 70k QA pairs for SFT and the 8k QA pairs for GRPO were created and what their specific attributes are. Additionally, since these datasets are used for model training and evaluation, it would be important to clarify whether there is any overlap between them to ensure the clarity and reliability of the experimental results.

---

> ### Author Rebuttal · Authors · 2025-07-30
>
> Thank you for the careful, constructive, and detailed review. We sincerely appreciate your recognition of the systematic methodology and practical value of our work. We have carefully reviewed your concerns and provide point-by-point responses below:
>
> ### **Q1: On dataset availability, construction process, and train-test split integrity.**
>
> **Open-sourcing Datasets**
>
> We **confirm** that we will **publicly release all resources** used in this paper, including the datasets **ShotBench and ShotQA**, model checkpoints, training hyperparameters, and evaluation code. All image and video samples in our datasets are obtained from **publicly accessible sources**, such as YouTube (for videos) and professional cinematography platforms (for images), and **each sample is directly downloadable via its corresponding URL**. To ensure transparency and long-term reproducibility, we will also release **detailed metadata** for each sample, including IMDb IDs, timestamps, and QA annotations, allowing users to trace or reconstruct the data even if source URLs become unavailable.
>
> **ShotQA Construction**
>
> ShotQA is constructed in a similar manner to ShotBench (Section 3.2), including:
>
> **Data filtering:** We apply quality filtering based on aesthetic scoring, content safety checks, and automated shot segmentation.
>
> **Annotator training:** Before labeling, annotators are trained using curated cinematography tutorials and domain-specific guidelines.
>
> **QA annotation:** Question–answer pairs are generated using structured templates and professional metadata.
>
> **Expert verification:** Annotations undergo multi-round expert review to ensure consistency and quality.
>
> Except that only video samples are totally manually annotated and verified by trained annotators and experts. For large-scale image annotations sourced from expert cinematography websites, we conducted random sampling checks and found their quality adequate for training use.
>
> **Dataset Splitting**
>
> We adopt a two-stage filtering strategy to ensure no overlap between training and evaluation sets:
>
> 1. We first remove duplicate samples based on IMDb IDs and timestamp as a coarse-level filtering step.
> 2. Then, we extract CLIP features for all samples and exclude samples from the training set whose feature has a cosine similarity greater than 0.95 with any sample in ShotBench.
>
> This strategy ensures no duplicate or near-duplicate samples are shared across splits. The 0.95 threshold is supported by existing works[1,2,3] and our own visual inspection. Additionally, we find that this choice allows us to retain samples that are visually similar yet differ in key cinematic attributes (usually adjacent shots within one scene),  such as shot size or composition, which are central to our task. This makes ShotBench more challenging and realistic.
>
> The GRPO sub-dataset consists of a combination of mid-difficulty samples and uniformly sampled QA pairs across all eight dimensions. We identify mid-difficulty samples by prompting Qwen2.5-VL-3B to answer a subset of the data multiple times and selecting those for which both correct and incorrect answers are observed across different runs. The remaining QA pairs were uniformly sampled again and used for SFT.
>
> We will add these details to Appendix in the revised version.
>
> ### **Q2: On copyright and data licensing concerns.**
>
>  All movie clips used in our dataset are sourced from **publicly available YouTube videos**, and we will provide:
>
> - Downloadable clips (in processed form)
> - Corresponding YouTube URLs
> - Metadata (title, timestamp, etc.)
> - Scripts for downloading and processing
>
> This ensures full transparency and reproducibility. We will also add a clarification in the paper to address potential copyright concerns directly and explicitly.
>
> ### **Q3: No comparison with existing cinematography understanding methods like SGNet.**
>
> We thank the reviewer for this suggestion. Unfortunately, **HunyuanVideo’s camera movement classifier** and **SGNet** are not open-sourced. Moreover, these methods use **limited and inconsistent taxonomies** that do not align with standard cinematography principles.
>
> To ensure fair comparison, we instead evaluated **CineScale** [4], an earlier open-source model for **shot size classification**. Note that CineScale only supports three coarse-grained categories (close-up, medium shot, wide shot). On the ShotBench shot size dimension, we tested CineScale in two settings:
>
> | Shot Size Samples                         | Accuracy (%) |
> | ----------------------------------------- | ------------ |
> | All samples                               | 21.2         |
> | Subset (close-up, medium shot, wide shot) | 51.2         |
>
> The overall performance is much lower than most existing VLMs (regard that Shot-VL achieves 82.5 accuracy in this dimension), which highlights both the difficulty of ShotBench and the limited transferability of prior shot classification methods.
>
> ### **Q4: How are multi-keyword answers evaluated?**
>
> All ShotBench questions are designed as four-option single-choice questions. For questions involving multiple keywords (e.g., lighting condition), each option contains the same number of keywords to maintain balance. Only the correct option includes all the correct keywords, while the distractors may contain a mixture of correct and incorrect keywords to enhance the challenge and maintain fairness. A prediction is considered correct only when the model selects the fully correct option. This design follows existing works[5,6,7] and provides a clear and consistent evaluation protocol.
>
> We will explicitly state this in the Evaluation section (Sec 3.3).
>
> ### **Q5: Have you tried applying your approach to larger models?**
>
> Yes, we have fine-tuned **Qwen2.5-VL-7B-Instruct** following the same two-stage training setting (SFT + GRPO), producing **ShotVL-7B**, it also achieves consistent improvements on ShotBench (**53.1 $\rightarrow$ 68.0**) and outperforms our previous ShotVL-3B (**65.1**).
>
> We are also conducting additional experiments on InternVL3-8B, and will report these results in the next version of the paper.
>
> ### **Formatting and grammar issues:**
>
> Thank you for the careful and helpful proofreading. We will comprehensively review the paper to correct all identified issues.
>
> ### References
>
> > [1] Gadre S Y, Ilharco G, Fang A, et al. Datacomp: In search of the next generation of multimodal datasets[J]. Advances in Neural Information Processing Systems, 2023, 36: 27092-27112.
> >
> > [2] Caron M, Iscen A, Fathi A, et al. A generative approach for wikipedia-scale visual entity recognition[C]//Proceedings of the IEEE/CVF Conference on Computer Vision and Pattern Recognition. 2024: 17313-17322.
> >
> > [3] Caron M, Fathi A, Schmid C, et al. Web-scale visual entity recognition: An LLM-driven data approach[J]. Advances in Neural Information Processing Systems, 2024, 37: 34533-34560.
> >
> > [4] Savardi M, Kovács A B, Signoroni A, et al. CineScale: A dataset of cinematic shot scale in movies[J]. Data in Brief, 2021, 36: 107002.
> >
> > [5] Chow W, Mao J, Li B, et al. Physbench: Benchmarking and enhancing vision-language models for physical world understanding[J]. arXiv preprint arXiv:2501.16411, 2025.
> >
> > [6] Fu C, Dai Y, Luo Y, et al. Video-mme: The first-ever comprehensive evaluation benchmark of multi-modal llms in video analysis[C]//Proceedings of the Computer Vision and Pattern Recognition Conference. 2025: 24108-24118.
> >
> > [7] Yue X, Ni Y, Zhang K, et al. Mmmu: A massive multi-discipline multimodal understanding and reasoning benchmark for expert agi[C]//Proceedings of the IEEE/CVF Conference on Computer Vision and Pattern Recognition. 2024: 9556-9567.

---

> ### Author Response · Authors · 2025-08-04
>
> Please let us know if there are any remaining questions or suggestions, we sincerely value your feedback.

---

### Official Review · Reviewer_K6tX · 2025-07-01

**Clarity:** 3
**Significance:** 3
**Originality:** 3
**Rating:** 5
**Confidence:** 4

**Summary:**

This paper introduces ShotBench, a benchmark designed to evaluate the ability of Vision-Language Models (VLMs) to comprehend the visual grammar of movie shots. ShotBench comprises 3,049 still images and 500 video clips from over 200 films, with each sample annotated by trained annotators or sourced from professional cinematography resources, resulting in 3,608 high-quality question-answer pairs. The authors conducted a comprehensive evaluation of over 20 state-of-the-art VLMs across eight core cinematography dimensions and found significant limitations in their fine-grained perception and cinematic reasoning. To enhance VLMs' capabilities in this area, the authors constructed the large-scale multimodal dataset ShotQA and proposed the ShotVL model, which outperformed existing models and set a new state-of-the-art performance.

**Questions:**

The detailed questions are outlined in the weaknesses.

**Ethical Concerns:**

["NO or VERY MINOR ethics concerns only"]

**Final Justification:**

The authors' rebuttal resolve most my conern.

**Limitations:**

Yes.

**Paper Formatting Concerns:**

None.

**Quality:**

3

**Strengths And Weaknesses:**

Strengths：

•	Innovative Benchmark and Dataset: The paper introduces ShotBench, the first benchmark specifically designed to evaluate Vision-Language Models (VLMs) on cinematography understanding, covering eight core dimensions (e.g., shot size, framing, lighting). It includes 3,608 high-quality question-answer pairs, addressing a critical gap in VLM evaluation for cinematic language.

•	Large-Scale Dataset: The ShotQA dataset, with approximately 70,000 QA pairs derived from movie shots, provides a valuable resource for training cinematography-aware VLMs.

•	Novel Model Proposal: The proposed ShotVL model, built on Qwen2.5-VL with supervised fine-tuning (SFT) and Group Relative Policy Optimization (GRPO), achieves a significant performance improvement (average 19.8% gain) on ShotBench, surpassing both open-source and proprietary models.

•	Comprehensive Evaluation: The evaluation of 24 leading VLMs provides a thorough analysis of current model limitations, offering clear directions for future research.

Weaknesses：

•	The determination of the eight core dimensions of cinematography primarily relies on Wikipedia, which may not be the most authoritative source. More authoritative classification methods could be considered, such as consulting industry standards, academic literature, or expert opinions in cinematography. This would enhance the credibility and robustness of the benchmark.

•	The dataset contains 3,049 still images and 500 video clips, which is not considered large-scale in the context of modern large models.

•	The paper does not provide detailed information on the data construction process, such as the number of annotators involved, the specific revision process, and the criteria for selecting and filtering data. A more detailed description of the data construction process, including the number of annotators, the training and oversight provided to them, and the specific steps taken to ensure data quality, would enhance transparency and reproducibility.

•	The baseline models used for comparison have not been specifically trained for cinematography understanding. This may affect the fairness of the comparison. Including an additional comparison experiment where the baseline models are fine-tuned on ShotBench would provide a more comprehensive evaluation. This would help to better understand the incremental improvements brought by the proposed methods and ensure a fair comparison. In addition, consider including DeepSeek's related VLM model in the comparison to provide a more comprehensive evaluation.

•	The failure cases shown in Figure 5 are not clearly attributed to specific models. It is unclear whether these failure cases are representative of all VLM models, including the proposed ShotVL model, or if they are specific to certain models. Specify which models the failure cases in Figure 5 represent. If these are common issues across all VLM models, including ShotVL, this should be explicitly stated. If the failure cases are specific to certain models, this should also be clearly indicated.

•	There are some instances of awkward phrasing and grammatical errors in the paper, which can detract from the overall readability and professionalism of the work.

---

> ### Author Rebuttal · Authors · 2025-07-30
>
> Thank you for the detailed and constructive review, as well as for the positive evaluation of our contributions. We have carefully reviewed the concerns and provide point-by-point responses below:
>
> ### **Q1: The eight core dimensions are primarily based on Wikipedia, which may not be the most authoritative source.**
>
> We appreciate the reviewer’s comment. To clarify, the Wikipedia page we referenced also cites a number of authoritative cinematography textbooks (e.g., *The Focal Dictionary of Photographic Technologies* [6], *The Art Direction Handbook for Film & Television* [7]), and our intention is to provide readers with a broad and accessible entry point into the topic. Our definitions are likewise grounded in established academic and professional sources. As detailed in Table 9, annotators were trained using materials from industry platforms like StudioBinder, and our taxonomy aligns with widely adopted cinematography textbooks, including Blain Brown’s *Cinematography: Theory and Practice* [1] and David Bordwell’s *Film Art: An Introduction* [2]. We will clarify this in the revised manuscript to enhance the credibility of our benchmark.
>
> ### **Q2: The dataset of ShotBench is not considered large-scale.**
>
> We apologize if the distinction between our datasets was unclear. To clarify:
>
> - **ShotQA** (~70k QA pairs) is our **large-scale dataset** used for training.
> - **ShotBench** (3.6k QA pairs) is our diagnostic evaluation benchmark.
>
> While ShotBench is compact by design, this reflects the typical nature of benchmarks, which prioritize **evaluation efficiency** and **diagnostic precision** over scale. ShotBench is fully annotated and verified by trained annotators and experts, making large-scale expansion costly but ensuring high reliability. Despite its size, it provides **diverse and comprehensive coverage** across the eight core dimensions of cinematography and associated terminology (as detailed in Table 7), enabling fine-grained model analysis. Moreover, its size is **comparable to or larger** than several widely used VLM benchmarks (e.g., MMVU [3], Video-MME [4]).
>
> We will clarify these roles and the rationale behind our design choices more explicitly in the revised manuscript.
>
> ### **Q3: Lack of detail on the data construction process**
>
> To clarify, we have provided a detailed data construction process in Section 3.2, including:
>
> **Data curation&filtering:** We apply datacuration and filtering based on aesthetic scoring, content safety checks, and automated shot segmentation.
>
> **Annotator training:** Before labeling, annotators are trained using curated cinematography tutorials and domain-specific guidelines.
>
> **QA annotation:** Question–answer pairs are generated using structured templates and professional annotations.
>
> **Expert verification:** Annotations undergo multi-round expert review to ensure consistency and quality.
>
> and **representative reference materials in Table 9**. In the revised version, we will expand our Appendix to include more detailed information on the data construction process, including the **aesthetic score threshold used for sample filtering (3.0)**, **the number of annotators involved (20)**, **additional materials and documentation** used during annotator training, and **more details on the expert review and validation** procedures.
>
> ### **Q4: Fairness of baseline comparison without fine-tuning and missing DeepSeek-VL models.**
>
> To clarify, our goal is to evaluate general-purpose VLMs under zero-shot settings to reflect their out-of-the-box understanding of cinematography. While fine-tuning all 24 baselines is infeasible, we agree it is valuable to test whether ShotQA improves other models. We have fine-tuned Qwen2.5-VL-7B on ShotQA and observed consistent gains (**53.1 $\rightarrow$​ 68.0**). Additionally, we evaluated the performance of DeepSeek-VL[5] on ShotBench, and we found that **DeepSeek-VL models perform poorly** on ShotBench (**35.0** for DeepSeek-VL2-tiny and **36.0** for DeepSeek-VL2-small).
>
> We will include these results in the next revision of the paper to provide a more comprehensive and fair comparison.
>
> ### **Q5: On the attribution of failure cases in Figure 5.**
>
> Thank you for highlighting this ambiguity. All failure cases shown in Figure 5 were **produced by GPT-4o**, the strongest proprietary model in our evaluation. We will revise the caption and discussion to clearly indicate the source model. Additionally, we will include failure cases from other representative models to offer a more balanced view of common challenges across models.
>
> ### **Q6: Language and grammar issues.**
>
> We sincerely appreciate your careful reading. We will thoroughly proofread the manuscript and correct all instances of awkward phrasing, grammatical issues, and unclear expressions to improve overall clarity and professionalism.
>
> ### References
>
> > [1] Brown B. Cinematography: theory and practice: image making for cinematographers and directors[M]. Routledge, 2016.
> >
> > [2] Bordwell D, Thompson K, Smith J. Film art: An introduction[M]. New York: McGraw-Hill, 2004.
> >
> > [3] Zhao Y, Zhang H, Xie L, et al. Mmvu: Measuring expert-level multi-discipline video understanding[C]//Proceedings of the Computer Vision and Pattern Recognition Conference. 2025: 8475-8489.
> >
> > [4] Fu C, Dai Y, Luo Y, et al. Video-mme: The first-ever comprehensive evaluation benchmark of multi-modal llms in video analysis[C]//Proceedings of the Computer Vision and Pattern Recognition Conference. 2025: 24108-24118.
> >
> > [5] Wu Z, Chen X, Pan Z, et al. Deepseek-vl2: Mixture-of-experts vision-language models for advanced multimodal understanding[J]. arXiv preprint arXiv:2412.10302, 2024.
> >
> > [6] Spencer D A. The Focal dictionary of photographic technologies[J]. London: Focal Press, 1973.
> >
> > [7] Rizzo M. The art direction handbook for film & television[M]. Routledge, 2014.

---

> > ### Comment · Reviewer_K6tX · 2025-08-03
> > **About the dataset quality**
> >
> > The authors mention that "ShotBench is compact by design." I'm curious about how the information density of each data clip influences the performance of MLLMs. Does this imply that we can achieve competitive results by training MLLMs on smaller but higher-quality datasets?

---

> ### Author Response · Authors · 2025-08-03
> **Response to the comment about dataset quality**
>
> Thank you for the follow-up question and your interest in the data quality aspect of our work. To clarify, ShotBench refers to our evaluation set and it's compact by design. In contrast, ShotQA is our training dataset (∼70k QA pairs). **We agree that data quality outweighs scale.** During data collection, we found many online images and videos mislabeled with noisy or inaccurate cinematography terms. Our early experiments showed that scaling with such data did not improve performance. We believe similar low-quality supervision exists in current MLLM training corpora, which is a key factor behind their limited performance in cinematic language understanding.

---

> ### Author Response · Authors · 2025-08-08
>
> Dear Reviewer K6tX,
>
> Thank you again for your detailed review and for engaging with us after the rebuttal.
>
> We’d greatly appreciate knowing whether our responses have addressed all of your concerns. If any concerns remain, we’d be happy to address them.
>
> We sincerely value your time and feedback.

---

### Official Review · Reviewer_S5g8 · 2025-07-02

**Clarity:** 4
**Significance:** 3
**Originality:** 4
**Rating:** 4
**Confidence:** 4

**Summary:**

This paper introduces a new benchmark (ShotBench) specifically designed to evaluate how well VLMs understand the visual grammar and cinematic language of movie shots. The authors evaluate 24 sota VLMs on this curated benchmark and find that existing models share common limitations in cinematic understanding. To address these gaps, the authors construct a 70k QA dataset (ShotQA), and use it to train a new model (ShotVL). Experimental results demonstrate that ShotQA significantly enhances ShotVL's performance, enabling it to surpass all previously evaluated open-source and proprietary models on ShotBench and establish a new sota.

**Questions:**

Please refer to the weaknesses above.

**Ethical Concerns:**

["NO or VERY MINOR ethics concerns only"]

**Final Justification:**

The authors' rebuttal has addressed my concerns. I am maintaining my rating of borderline accept.

**Limitations:**

yes

**Quality:**

3

**Strengths And Weaknesses:**

### Strengths

* While prior work has addressed specific sub-tasks in film understanding (e.g., shot classification), this work has its unique novelty in proposing a comprehensive and structured evaluation of cinematography understanding since the proposed benchmark covers eight core aspects of cinematography (shot size, shot framing, camera angle, lens size, lighting type, lighting condition, composition, and camera movement).

* This work presents extensive experiments on the proposed benchmark, evaluating 24 widely used VLMs from both open-source and close-source, and provides detailed quantitative results along with insightful qualitative analyses. The inclusion of annotated image examples and failure cases is effective in conveying the subtlety of the task and the types of errors current models make.

* The proposed model (ShotVL) is trained with a two-stage pipeline combining supervised fine-tuning and GRPO on the proposed QA dataset (ShotQA). The resulting model achieves substantial improvements over strong baselines, including GPT-4o, which is a clear evidence to verify the effectiveness of the proposed dataset.


### Weaknesses

* One concern regarding the proposed benchmark/dataset is that, the QA pairs are provided in the form on multiple-choice. It is unclear how well the benchmark generalizes to more open-ended or generative tasks in cinematic language understanding. In addition, such task design may not fully capture the reasoning capabilities of tested VLMs that perform better on open-ended or generation tasks.

* This is not a major concern, but it would strengthen the claims to show that ShotQA and GRPO training strategies is model-agnostic and may generalize across different VLM architectures of various sizes. Currently, the proposed ShotVL only use one relatively small VLM as backbone (Qwen2.5-VL-3B-Instruct).

---

> ### Author Rebuttal · Authors · 2025-07-30
>
> Thank you for your positive assessment and for highlighting the novelty and extensive experiments of our work. We sincerely appreciate your valuable feedback and constructive suggestions. We have carefully reviewed your concerns and provide point-by-point responses below:
>
> ### **Q1: The multiple-choice (MCQ) format limits the evaluation of generative capabilities.**
>
> We agree that open-ended evaluation is an important direction for cinematic understanding. We adopt a multiple-choice format in ShotBench to ensure **objectivity**, **scalability**, and **reproducibility**, which are key qualities for building a reliable benchmark in a new domain. This design follows the common practice in recent representive VLM benchmarks (e.g., Video-MME [1], PhysBench [2], MMBench [3]) and enables fair comparison across models. While it does not capture generative reasoning, we view it as a necessary first step and will clarify this limitation in the revised version.
>
> ### **Q2: Strengthen the claims to show that ShotQA and GRPO training strategies is model-agnostic and may generalize across different VLM architectures of various sizes.**
>
> We appreciate the reviewer’s suggestion. In response, we have conducted additional experiments using a stronger backbone, Qwen2.5-VL-7B-Instruct, following the same two-stage training setting (SFT + GRPO). This larger model, **ShotVL-7B**, achieves better overall performance (53.1 $\rightarrow$ **68.0**) than ShotVL-3B (**65.1**) on ShotBench.
>
> We will include these results, along with more extended comparisons and analysis, in the next revision of the paper.
>
> ### References
>
> > [1] Fu C, Dai Y, Luo Y, et al. Video-mme: The first-ever comprehensive evaluation benchmark of multi-modal llms in video analysis[C]//Proceedings of the Computer Vision and Pattern Recognition Conference. 2025: 24108-24118.
> >
> > [2] Chow W, Mao J, Li B, et al. Physbench: Benchmarking and enhancing vision-language models for physical world understanding[J]. arXiv preprint arXiv:2501.16411, 2025.
> >
> > [3] Liu Y, Duan H, Zhang Y, et al. Mmbench: Is your multi-modal model an all-around player?[C]//European conference on computer vision. Cham: Springer Nature Switzerland, 2024: 216-233.

---

> ### Author Response · Authors · 2025-08-04
>
> Please let us know if there are any remaining questions or suggestions, we sincerely value your feedback.

---

> > ### Comment · Reviewer_S5g8 · 2025-08-05
> >
> > Thanks the authors for the rebuttal response, which has addressed my concerns. I am maintaining my rating of borderline accept.

---

### Decision · Program_Chairs · 2025-09-17

**Decision:**

Accept (poster)

**Comment:**

Summary

This paper introduces ShotBench, a new benchmark for evaluating Vision-Language Models (VLMs) on their understanding of cinematic language, covering eight core dimensions of cinematography. The authors curate a high-quality dataset (ShotBench: 3,608 QA pairs from 3,049 images and 500 video clips, annotated by experts) and conduct a comprehensive evaluation of 24 state-of-the-art VLMs, revealing significant limitations in fine-grained cinematic reasoning. To address these gaps, they construct ShotQA (∼70k QA pairs) and propose ShotVL, a VLM trained with supervised fine-tuning and Group Relative Policy Optimization (GRPO). ShotVL achieves substantial improvements over both open-source and proprietary baselines, establishing new state-of-the-art results on ShotBench.

Strengths
- Novelty and Scope: First benchmark and dataset specifically targeting cinematic language understanding in VLMs, covering eight well-defined dimensions (shot size, framing, camera angle, lens size, lighting type, lighting condition, composition, camera movement).
- Comprehensive Evaluation: Rigorous analysis of 24 VLMs, including both open-source and proprietary models, with detailed quantitative and qualitative results.
- High-Quality Annotation: Expert-validated annotations and a transparent, reproducible data construction process.
- Model and Training Innovations: Introduction of ShotVL, trained with a two-stage pipeline (SFT + GRPO), demonstrating clear performance gains.
- Resource Commitment: Authors commit to open-sourcing all datasets, code, and metadata, supporting reproducibility and future research.

Weaknesses
- Benchmark Format: The multiple-choice QA format, while objective and scalable, may not fully capture open-ended or generative reasoning capabilities of VLMs.
- Dataset Size and Construction Details: ShotBench is compact by design; some reviewers questioned whether its scale is sufficient for modern models. Initial manuscript lacked detail on annotation process, inter-annotator agreement, and data splits, though these were clarified in the rebuttal.
- Taxonomy Source: The eight dimensions were initially based on Wikipedia; reviewers suggested grounding in more authoritative academic or industry sources, which the authors addressed by referencing standard textbooks and professional materials.
- Baseline Comparisons: Some baselines were not fine-tuned for cinematography understanding, raising questions about fairness. Authors added new experiments with fine-tuned models and additional baselines (e.g., DeepSeek-VL).

This submission makes an important contribution to the evaluation and advancement of multimodal models in a domain (cinematography) that is both technically challenging and of growing practical importance, especially with the rise of text to video models. The benchmark, dataset, and model are novel, and the authors have demonstrated thoroughness in both evaluation and addressing reviewer concerns.